# LATENT HIERARCHICAL IMITATION LEARNING FOR STOCHASTIC ENVIRONMENTS

## ABSTRACT

Many applications of imitation learning require the agent to avoid mode collapse and mirror the full distribution of observed behaviours. Existing methods that address this *distributional realism* typically rely on hierarchical policies conditioned on sampled *types* that model agent-internal features like persona, goal, or strategy. However, these methods are often inappropriate for stochastic environments, where internal and external factors of influence on the observed agent trajectories have to be disentangled, and only internal factors should be encoded in the agent type to be robust to changing environment conditions. We formalize this challenge as distribution shift in the *conditional* distribution of agent types under environmental stochasticity, in addition to the familiar covariate shift in state visitations. We propose *Robust Type Conditioning* (`RTC`), which eliminate this shifts with adversarial training under randomly sampled types. Experiments on two domains, including the large-scale *Waymo Open Motion Dataset*, show improved distributional realism while maintaining or improving task performance compared to state-of-the-art baselines.

## 1 INTRODUCTION

Learning to imitate behaviour is crucial when reward design is infeasible (Amodei et al., 2016; Hadfield-Menell et al., 2017; Fu et al., 2018; Everitt et al., 2021), for overcoming hard exploration problems (Rajeswaran et al., 2017; Zhu et al., 2018), and for realistic modelling of dynamical systems with multiple interacting agents (Farmer and Foley, 2009). Such systems, including games, driving simulations, and agent-based economic models, often have known state transition functions, but require accurate agents to be realistic. For example, for driving simulations, which are crucial for accelerating the development of autonomous vehicles (Suo et al., 2021; Igl et al., 2022), faithful reactions of all road users are paramount. Furthermore, it is not enough to mimic a single mode in the data; instead, agents must reproduce the full distribution of behaviours to avoid sim2real gaps in modelled systems (Grover et al., 2018; Liang et al., 2020), under-explored solutions in complex tasks (Vinyals et al., 2019) and suboptimal policies in games requiring mixed strategies (Nash Jr, 1950).

Current imitation learning (IL) methods fall short of achieving such *distributional realism* by matching all modes in the data. The required stochastic policy cannot be recovered from a fixed reward function and adversarial methods, while aiming to match the distribution in principle, are known to be prone to mode collapse in practice (Wang et al., 2017; Lucic et al., 2018; Creswell et al., 2018). Furthermore, progress on distributional realism is hindered by a lack of suitable IL benchmarks, with most relying on unimodal data and only evaluating task performance as measured by rewards, but not mode coverage. By contrast, many applications require distributional realism in addition to good task performance. For example, accurately evaluating the safety of autonomous vehicles in simulation relies on distributionally realistic agents. Consequently, our goal is to improve distributional realism while maintaining strong task performance.

To mitigate mode collapse in complex environments, previous work uses hierarchical policies in an auto-encoder framework (Wang et al., 2017; Suo et al., 2021; Igl et al., 2022). During training, an encoder infers

latent variables from observed trajectories and the agent, conditioned on those latent variables, strives to imitate the original trajectory. At test time, a prior distribution proposes distributionally realistic latent values, without requiring access to privileged future information. We refer to this latent vector as an agent's inferred *type* since it expresses intrinsic characteristics of the agent that yield the multimodal behaviour. Depending on the environment, the *type* could, for example, represent the agent's persona, belief, goal, or strategy.

However, these hierarchical methods rely on either manually designed type representations (Igl et al., 2022) or the strong assumption that *all* stochasticity in the environment can be controlled by the agent (Wang et al., 2017; Suo et al., 2021). Unfortunately, this assumption is violated in most realistic scenarios. For example, in the case of driving simulations, trajectories depend not only on the agent's type, expressing its driving style and intent, but also on external factors such as the behaviour of other road users. Crucially, despite being inferred from future trajectories during training, agent types must be independent of these external factors to avoid leaking information about future events outside the agent's control, which in turn can impair generalization at test time under changed, and ex-ante unknown, environmental conditions. In other words, the challenge in learning hierarchical policies using IL in stochastic environments is to disentangle the internal and external factors of influence on the trajectories and only encode the former into the type.

Consider the example of an expert approaching an intersection at the same time as another car. The expert passes if the other car brakes and yields to it otherwise. To reconstruct the scene with ease, a naively trained latent model could not only encode the agent's intended direction (an *internal* decision) but also whether to yield, which depends on the other car (an *external* factor). This is catastrophic at test time when the latent, and hence the yielding decision, is sampled independently of the other car's behaviour. In contrast, if only the expert's intent were encoded in the latent, the policy would learn to react appropriately to external factors.

In this paper, we identify these subtle challenges arising under stochastic environments and formulate them as a new form of distribution shift for hierarchical policies. Unlike the familiar covariate shift in the state distribution (Ross et al., 2011), this *conditional type shift* occur in the distribution of the inferred latent type. It greatly reduces performance by yielding causally confused agents that rely on the latent type for information about external factors, instead of inferring them from the latest environment observation. We propose *Robust Type Conditioning* (`RTC`) to eliminate this distribution shift and avoid causally confused agents through a coupled adversarial training objective under randomly sampled types. We do not require access to an expert, counterfactuals, or manually specified type labels for trajectories.

Experimentally, we show the need for improved distributional realism due to mode collapse in state-of-the-art imitation learning techniques such as GAIL (Ho and Ermon, 2016). Furthermore, we show that naively trained hierarchical models with inferred types improve distributional realism, but exhibit poor task performance in stochastic environments. By contrast, `RTC` can maintain good task performance in stochastic environments while improving distributional realism and mode coverage. We evaluate `RTC` on the illustrative *Double Goal Problem* as well as the large scale *Waymo Open Motion Dataset* (Ettinger et al., 2021) of real driving behaviour.

## 2 BACKGROUND

We are given a dataset $\mathcal{D} = \{\tau_i\}_{i=1}^{N}$ of $N$ trajectories $\tau_i = s_0^{(i)}, a_0^{(i)}, \ldots s_T^{(i)}$, drawn from $p(\tau)$ of one or more experts interacting with a stochastic environment $p(s_{t+1}|s_t, a_t)$ where $s_t \in \mathcal{S}$ are states and $a_t \in \mathcal{A}$ are actions. Our goal is to learn a policy $\pi_\theta(a_t|s_t)$ to match $p(\tau)$ when replacing the unknown expert and generating rollouts $\hat{\tau} \sim p(\hat{\tau}) = p(s_0) \prod_{t=0}^{T-1} \pi_\theta(\hat{a}_t|\hat{s}_t) p(\hat{s}_{t+1}|\hat{s}_t, \hat{a}_t)$ from the inital states $s_0 \sim p(s_0)$. We simplify notation and write $\hat{\tau} \sim \pi_\theta(\hat{\tau})$ and $\tau \sim \mathcal{D}(\tau)$ to indicate rollouts generated by the policy or drawn from the data respectively. Expectations $\mathbb{E}_{\tau \sim \mathcal{D}}$ and $\mathbb{E}_{\hat{\tau} \sim \pi_\theta}$ are taken over all pairs $(s_t, a_t) \in \tau$ and $(\hat{s}_t, \hat{a}_t) \in \hat{\tau}$.

Previous work (e.g., Ross et al., 2011; Ho and Ermon, 2016) shows that a core challenge of learning from demonstration is reducing or eliminating the covariate shift in the state-visitation frequencies $p(s)$ caused by accumulating errors when using $\pi_\theta$. Unfortunately, *Behavioural Cloning* (BC), a simple supervised training objective optimising $\max_\theta \mathbb{E}_{\tau \sim \mathcal{D}} [\log \pi_\theta(a_t|s_t)]$ is not robust to it. To overcome covariate shift, generative

adversarial imitation learning (GAIL) (Ho and Ermon, 2016) optimises $\boldsymbol{\pi_\theta}$ to fool a learned discriminator $D_\phi(\hat{\boldsymbol{a}}_t, \hat{\boldsymbol{s}}_t)$ that is trained to distinguish between trajectories in $\mathcal{D}$ and those generated by $\boldsymbol{\pi_\theta}$:

$$\min_{\boldsymbol{\theta}} \max_{\boldsymbol{\phi}} \mathbb{E}_{\hat{\tau} \sim \boldsymbol{\pi_\theta}} \Big[ \log(D_\phi(\hat{\boldsymbol{a}}_t, \hat{\boldsymbol{s}}_t)) \Big] + \mathbb{E}_{\tau \sim \mathcal{D}} \Big[ \log(1 - D_\phi(\boldsymbol{a}_t, \boldsymbol{s}_t)) \Big]. \tag{1}$$

The policy can be optimised using reinforcement learning, by treating the log-discriminator scores as costs, $r_t = -\log D_\phi(\hat{\boldsymbol{a}}_t, \hat{\boldsymbol{s}}_t)$. Alternatively, if the policy can be reparameterized (Kingma and Welling, 2013) and the environment is differentiable, the sum of log discriminator scores can be optimised directly without relying on high-variance score function estimators by backpropagating through the transition dynamics, $\mathcal{L}_{\text{adv}}(\hat{\tau}) = \mathbb{E}_{\hat{\tau} \sim \boldsymbol{\pi_\theta}} [\sum_t -\log D_\phi(\hat{\boldsymbol{a}}_t, \hat{\boldsymbol{s}}_t)]$. We refer to this as *Model-based GAIL* (MGAIL), though in contrast to Baram et al. (2016), we assume a known differentiable environment instead of a learned model.

In this work, we are concerned with multimodal distributions $p(\tau)$ and how mode collapse can be avoided when learning $\boldsymbol{\pi_\theta}$. To this end, we assume the dataset is sampled from $p(\tau) = p(\boldsymbol{s}_0) \int p(\boldsymbol{g})p(\boldsymbol{\xi}) \prod_{t=0}^{T} p(\boldsymbol{a}_t|\boldsymbol{s}_t, \boldsymbol{g})p(\boldsymbol{s}_{t+1}|\boldsymbol{s}_t, \boldsymbol{a}_t, \boldsymbol{\xi})d\boldsymbol{\xi}d\boldsymbol{g}$, where $\boldsymbol{g}$ is the agent *type*, expressing agent characteristics such as persona, goal, or, strategy, and $\boldsymbol{\xi}$ is a random variable capturing the stochasticity in the environment, i.e. $p(\boldsymbol{s}_{t+1}|\boldsymbol{s}_t, \boldsymbol{a}_t, \boldsymbol{\xi})$ is a delta distribution $\delta_{f(\boldsymbol{s}_t, \boldsymbol{a}_t, \boldsymbol{\xi})}(\boldsymbol{s}_{t+1})$ for some transition function $f$. Learned agents matching $p(\tau)$, i.e., with $p(\hat{\tau}) \approx p(\tau)$, are *distributionally realistic*, whereas *realism* describes single trajectories when $\hat{\tau}$ lies in the support of $p_\tau(\tau)$. As we show in section 6, current non-hierarchical adversarial methods (Ho and Ermon, 2016) exhibit mode collapse and are not distributionally realistic.

To combat mode collapse, hierarchical methods (e.g., Wang et al., 2017; Lynch et al., 2020; Suo et al., 2021; Igl et al., 2022) often rely on an encoder to infer latent agent types $\hat{\boldsymbol{g}}_e$ from trajectories during training, $\hat{\boldsymbol{g}}_e \sim \boldsymbol{e_\theta}(\hat{\boldsymbol{g}}_e|\tau)$, and optimise the control policy $\boldsymbol{\pi_\theta}(\hat{\boldsymbol{a}}_t|\hat{\boldsymbol{s}}_t, \hat{\boldsymbol{g}}_e)$ to generate trajectories $\hat{\tau}_e$ similar to $\tau$: $\hat{\tau}_e \sim p(\hat{\tau}_e|\hat{\boldsymbol{g}}_e) = p(\boldsymbol{s}_0) \prod_{t=0}^{T-1} \boldsymbol{\pi_\theta}(\hat{\boldsymbol{a}}_t|\hat{\boldsymbol{s}}_t, \hat{\boldsymbol{g}}_e)p(\hat{\boldsymbol{s}}_{t+1}|\hat{\boldsymbol{a}}_t, \hat{\boldsymbol{s}}_t)$, with $\hat{\boldsymbol{g}}_e \sim \boldsymbol{e_\theta}(\hat{\boldsymbol{g}}_e|\tau)$. If ground truth trajectories are not accessible during testing, a prior $p_{\boldsymbol{\theta}}(\hat{\boldsymbol{g}}_p)$ can be used to sample distributionally realistic types $\hat{\boldsymbol{g}}_p$. We indicate by subscript $\hat{\boldsymbol{g}}_p$ or $\hat{\boldsymbol{g}}_e$ whether the inferred type and trajectory are drawn from the prior distribution $p_{\boldsymbol{\theta}}(\hat{\boldsymbol{g}}_p)$ or encoder $\boldsymbol{e_\theta}(\hat{\boldsymbol{g}}_e|\tau)$. Subscripts are omitted for states and actions to simplify notation. Inferred types and predicted trajectories without subscripts indicate that either sampling distribution could be used. For discussing information theoretic quantities, we will use capital letters $\boldsymbol{S}, \boldsymbol{A}, \hat{\boldsymbol{A}}, \hat{\boldsymbol{G}}$ and $\Xi$ to denote the random variables for values $\boldsymbol{s}, \boldsymbol{a}, \hat{\boldsymbol{a}}, \hat{\boldsymbol{g}}$ and $\boldsymbol{\xi}$.

## 3 CONDITIONAL TYPE SHIFT IN STOCHASTIC ENVIRONMENTS

In this section we outline a challenge that arises for hierarchical policies in stochastic environments. A shift in the conditional type distribution can arise because latent types are drawn from the encoder $\boldsymbol{e_\theta}(\hat{\boldsymbol{g}}_e|\tau)$ during training but from the prior $p_{\boldsymbol{\theta}}(\hat{\boldsymbol{g}}_p)$ during testing. While the prior is trained to match the *marginal* distribution of the encoder $p(\hat{\boldsymbol{g}}_e) = \mathbb{E}_{\boldsymbol{g},\boldsymbol{\xi}} [\boldsymbol{e_\theta}(\hat{\boldsymbol{g}}_e|\tau)]$, this is not the case for the *conditional* distribution $p(\hat{\boldsymbol{g}}_e|\boldsymbol{\xi}) = \mathbb{E}_{\boldsymbol{g}} [\boldsymbol{e_\theta}(\hat{\boldsymbol{g}}_e|\tau)]$. We show that this conditional type shift can result in policies ignoring environmental information. In section 6 we experimentally confirm that this translates to reduced task performance.

We use the simplified model in fig. 1 to describe the consequences of the conditional type shift. This model has two sources of randomness in the data $\mathcal{D}$: the environmental noise $\boldsymbol{\xi}$ and the multimodal type $\boldsymbol{g}$ of the expert we are mimicking. The crucial difference between $\boldsymbol{g}$ and $\boldsymbol{\xi}$ is that $\boldsymbol{\xi}$ represents *external* factors that the agent cannot control but to which it has to react, while $\boldsymbol{g}$ encodes agent-*internal* decisions that can be taken independently of $\boldsymbol{\xi}$. In this simplified model, the state $\boldsymbol{s}$ is a deterministic function of only $\boldsymbol{\xi}$ as we disregard cross-temporal dependencies. During training, when real trajectories $\tau = (\boldsymbol{s}, \boldsymbol{a})$ are available, the inferred type $\hat{\boldsymbol{g}}_e$ is drawn from the encoder $\boldsymbol{e_\theta}(\hat{\boldsymbol{g}}_e|\tau)$. During testing, without access to $\tau$, a prior $p_{\boldsymbol{\theta}}(\hat{\boldsymbol{g}}_p)$ is used. Actions $\hat{\boldsymbol{a}}$ are drawn from the learned control policy $\boldsymbol{\pi_\theta}(\hat{\boldsymbol{a}}|\boldsymbol{s}, \hat{\boldsymbol{g}})$ and optimisation is performed to minimise a reconstruction loss $\mathcal{L}_{\text{rec}}(\boldsymbol{a}, \hat{\boldsymbol{a}})$. We express the core result of the policy 'ignoring environmental information' using mutual information $I(\boldsymbol{S}, \boldsymbol{A})$, entropy $H(\boldsymbol{A}, \hat{\boldsymbol{G}}_e)$ and conditional entropy $H(\boldsymbol{A}|\hat{\boldsymbol{G}}_e)$.

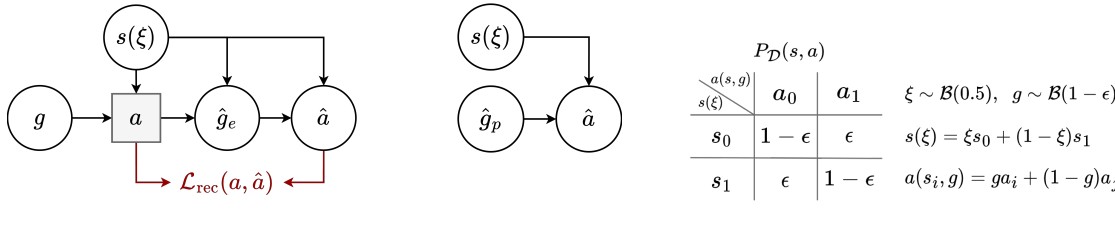

**(a)** Encoder: $\hat{\boldsymbol{g}}_e \sim \boldsymbol{e_\theta}(\hat{\boldsymbol{g}}_e|\tau)$    **(b)** Prior: $\hat{\boldsymbol{g}}_p \sim p_{\boldsymbol{\theta}}(\hat{\boldsymbol{g}}_p)$.    **(c)** Example dataset.

**Figure 1:** Simplified, non-temporal setup with environmental noise $\boldsymbol{\xi}$ and multi-modality induced by the unobserved agent type $\boldsymbol{g}$. We denote $\tau = (\boldsymbol{s}, \boldsymbol{a})$. The inferred type $\hat{\boldsymbol{g}}$ is sampled from $\boldsymbol{e_\theta}(\hat{\boldsymbol{g}}_e|\tau)$ during training (*left*) and $p_{\boldsymbol{\theta}}(\hat{\boldsymbol{g}}_p)$ otherwise (*middle*). The control policy is $\boldsymbol{\pi_\theta}(\hat{\boldsymbol{a}}|\boldsymbol{s}, \hat{\boldsymbol{g}})$. Circles are random variables and squares deterministic functions. The loss $\mathcal{L}(\boldsymbol{a}, \hat{\boldsymbol{a}})$ penalises differences between $\boldsymbol{a}$ and $\hat{\boldsymbol{a}}$. *Right:* Example data, $\mathcal{B}$ denotes Bernoulli distributions.

**Theorem 1.** We assume the model $p_{\boldsymbol{\theta}}(\hat{\boldsymbol{a}}|\boldsymbol{s}, \boldsymbol{a}) = \int \boldsymbol{e_\theta}(\hat{\boldsymbol{g}}_e|\boldsymbol{a}, \boldsymbol{s})\boldsymbol{\pi_\theta}(\hat{\boldsymbol{a}}|\boldsymbol{s}, \hat{\boldsymbol{g}}_e)d\hat{\boldsymbol{g}}_e$ is achieving optimal reconstruction loss $\mathcal{L}_{\text{rec}} = 0$ on $P_{\mathcal{D}}(\boldsymbol{s}, \boldsymbol{a})$. The test policy is $p_{\boldsymbol{\theta}}(\hat{\boldsymbol{a}}|\boldsymbol{s}) = \int p_{\boldsymbol{\theta}}(\hat{\boldsymbol{g}}_p)\boldsymbol{\pi_\theta}(\hat{\boldsymbol{a}}|\boldsymbol{s}, \hat{\boldsymbol{g}}_p)d\hat{\boldsymbol{g}}_p$ with the marginal encoder $p_{\boldsymbol{\theta}}(\hat{\boldsymbol{g}}) = \mathbb{E}_{P_{\mathcal{D}}}[\boldsymbol{e_\theta}(\hat{\boldsymbol{g}}|\boldsymbol{a}, \boldsymbol{s})]$ as prior distribution. We can say for the training distribution $P(\boldsymbol{s}, \boldsymbol{a}, \hat{\boldsymbol{g}}_e) = P_{\mathcal{D}}(\boldsymbol{s}, \boldsymbol{a})\boldsymbol{e_\theta}(\hat{\boldsymbol{g}}_e|\boldsymbol{s}, \boldsymbol{a})$ and and testing distribution $P(\boldsymbol{s}, \hat{\boldsymbol{a}}, \hat{\boldsymbol{g}}_p) = P_{\mathcal{D}}(\boldsymbol{s})p_{\boldsymbol{\theta}}(\hat{\boldsymbol{g}}_p)\boldsymbol{\pi_\theta}(\hat{\boldsymbol{a}}|\boldsymbol{s}, \hat{\boldsymbol{g}}_p)$: If $H(\boldsymbol{A}|\hat{\boldsymbol{G}}_e) < I(\boldsymbol{S}, \boldsymbol{A})$ and $H(\boldsymbol{A}, \hat{\boldsymbol{G}}_e) = H(\hat{\boldsymbol{A}}, \hat{\boldsymbol{G}}_p)$, then $I(\boldsymbol{S}, \hat{\boldsymbol{A}}) < I(\boldsymbol{S}, \boldsymbol{A})$.

**Corollary 1.** If $H(\boldsymbol{A}|\hat{\boldsymbol{G}}_e) = 0$, the assumption $H(\boldsymbol{A}, \hat{\boldsymbol{G}}_e) = H(\hat{\boldsymbol{A}}, \hat{\boldsymbol{G}}_p)$ becomes unnecessary in theorem 1 and we have $I(\boldsymbol{S}, \hat{\boldsymbol{A}}) = 0 < I(\boldsymbol{S}, \boldsymbol{A})$.

Proofs are in appendix A. The core result is that the mutual information between states and actions is lower for prior policies than in the data, implying that such policies ignore action-relevant information in the states. This happens if $H(\boldsymbol{A}|\hat{\boldsymbol{G}}_e) < I(\boldsymbol{S}, \boldsymbol{A})$, i.e. if the type $\hat{\boldsymbol{g}}$ captures too much information about $\hat{\boldsymbol{a}}$ during training. The condition $H(\boldsymbol{A}, \hat{\boldsymbol{G}}_e) = H(\hat{\boldsymbol{A}}, \hat{\boldsymbol{G}}_p)$ assures that the entropy in the system and mutual information between variables remains comparable between training and testing. In the extreme case that type fully determines the action, i.e. $H(\boldsymbol{A}|\hat{\boldsymbol{G}}_e) = 0$, the policy ignores the state entirely, i.e. $I(\boldsymbol{S}, \hat{\boldsymbol{A}}) = 0$.

Because the encoder and policy are trained jointly, the failure case $H(\boldsymbol{A}|\hat{\boldsymbol{G}}_e) < I(\boldsymbol{S}, \boldsymbol{A})$ requires the encoder to capture too much information about $\boldsymbol{s}$ in $\hat{\boldsymbol{g}}_e$ *and* the policy relying too much on $\hat{\boldsymbol{g}}_e$ to predict $\boldsymbol{a}$, which constitutes a form of *causal confusion*. Without the encoder providing excessive information about $\boldsymbol{s}$, the policy could not learn to over-rely on $\hat{\boldsymbol{g}}_e$. Conversely, even with excessive information about $\boldsymbol{s}$ in $\hat{\boldsymbol{g}}_e$, the policy could still ignore it and avoid $H(\boldsymbol{A}|\hat{\boldsymbol{G}}_e) < I(\boldsymbol{S}, \boldsymbol{A})$.

As an example, for the data given in fig. 1c, this is a solution satisfying $I(\boldsymbol{S}, \boldsymbol{A}) > H(\hat{\boldsymbol{A}}|\hat{\boldsymbol{G}}) = 0$:

$$\hat{\boldsymbol{g}}_e(\boldsymbol{s}_i, \boldsymbol{a}_j) = \begin{cases} 0 & \text{if } j = 0 \\ 1 & \text{if } j = 1 \end{cases} \quad \text{implies} \quad \boldsymbol{\pi_\theta}(\hat{\boldsymbol{a}}|\boldsymbol{s}_i, \hat{\boldsymbol{g}}) = \begin{cases} \boldsymbol{a}_0 & \text{if } \hat{\boldsymbol{g}} = 0 \\ \boldsymbol{a}_1 & \text{if } \hat{\boldsymbol{g}} = 1 \end{cases} \quad \text{and} \quad p_{\boldsymbol{\theta}}(\hat{\boldsymbol{g}}_p) = \mathcal{B}(0.5). \quad (2)$$

with Bernoulli distribution $\mathcal{B}$. The latent type fully determines the action and the policy ignores the state. This allows perfect reconstruction during training, but fails at test time when $p_{\boldsymbol{\theta}}(\hat{\boldsymbol{g}}_p)\boldsymbol{\pi_\theta}(\hat{\boldsymbol{a}}|\boldsymbol{s}, \hat{\boldsymbol{g}}_p)$ would randomly sample $\boldsymbol{a}_0$ and $\boldsymbol{a}_1$ with equal probability. In the example of a car approaching an intersection, this corresponds to entering the intersection independently of whether another car is approaching quickly, leading to increased collision rates. Also note that the conditional type distribution is not independent of the state, i.e. $p(\hat{\boldsymbol{g}}_e|\boldsymbol{s}_0) = \mathcal{B}(1 - \epsilon) \neq p(\hat{\boldsymbol{g}}_e|\boldsymbol{s}_1) = \mathcal{B}(\epsilon)$, leading to a conditional type shift to $p_{\boldsymbol{\theta}}(\hat{\boldsymbol{g}}_p) = \mathcal{B}(0.5)$ at test time.

Other training solutions exist for which $I(\boldsymbol{S}, \boldsymbol{A}) = H(\hat{\boldsymbol{A}}|\hat{\boldsymbol{G}})$ and $I(\boldsymbol{S}, \hat{\boldsymbol{A}}) = I(\boldsymbol{S}, \boldsymbol{A})$. For example:

$$\hat{\boldsymbol{g}}_e(\boldsymbol{s}_i, \boldsymbol{a}_j) = \begin{cases} 0 & \text{if } i = j \\ 1 & \text{if } i \neq j \end{cases}, \quad \boldsymbol{\pi_\theta}(\hat{\boldsymbol{a}}|\boldsymbol{s}_i, \hat{\boldsymbol{g}}) = \begin{cases} \boldsymbol{a}_i & \text{if } \hat{\boldsymbol{g}} = 0 \\ \boldsymbol{a}_j & \text{if } \hat{\boldsymbol{g}} = 1 \end{cases} \text{ for } i \neq j, \quad p_{\boldsymbol{\theta}}(\hat{\boldsymbol{g}}_p) = \mathcal{B}(1 - \epsilon). \quad (3)$$

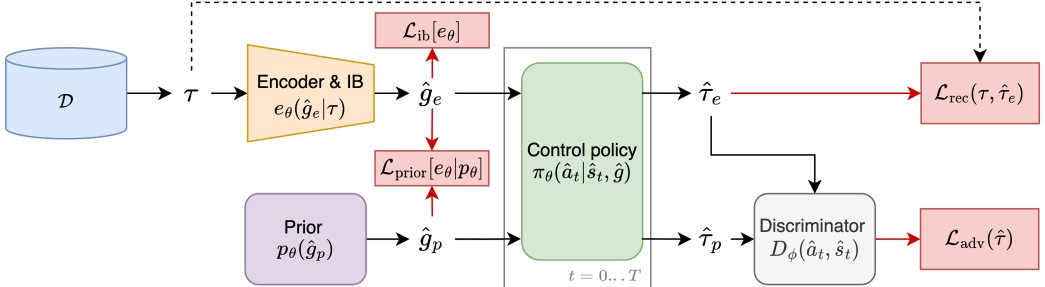

**Figure 2:** *Robust Type Conditioning* (RTC): The control policy $\pi_\theta(\hat{a}_t|\hat{s}_t, \hat{g})$ is trained under inferred types $\hat{g}$ sampled from both the encoder $e_\theta(\hat{g}_e|\tau)$ and the prior $p_\theta(\hat{g}_p)$. The reconstruction loss $\mathcal{L}_{\text{rec}}(\tau, \hat{\tau}_e)$ avoids mode collapse. The adversarial loss $\mathcal{L}_{\text{adv}}(\hat{\tau}_p)$ under prior types prevents causally confused policies and ensures good task performance. $\mathcal{L}_{\text{prior}}$ optimises the prior to sample distributionally realistic types and the information bottleneck loss $\mathcal{L}_{\text{ib}}$ reduces covariate shift.

Here the latent type $\hat{g}$ only captures the agent-internal randomness, the conditional type distribution matches the prior, i.e. $p(\hat{g}_e|s_1) = p(\hat{g}_e|s_2) = p_\theta(\hat{g}_p) = \mathcal{B}(1 - \epsilon)$, and the test policy correctly reproduces the data.

For temporally extended data, the states $s_t$ will depend not only on $\xi$, but also on $g$ or $\hat{g}$, complicating theoretical treatment. Nevertheless, seeing $\xi$ as all *future* stochasticity in the environment, the same threat of conditional type shift arises. In the next section, we introduce two training interventions. The first discourages causally confused policies, the other discourages the encoder from capturing excessive information about $\xi$. In section 6 we show that using both interventions jointly allows to use hierarchical policies for improved distributional realism while avoiding the sub-optimal solution for which $I(S, A) > I(S, \hat{A})$.

## 4 ROBUST TYPE CONDITIONING

We present *Robust Type Conditioning* (RTC), a method for improving distributional realism in imitation learning while maintaining high task performance. RTC follows the auto-encoder framework discussed in sections 2 and 3 but avoids conditional type shifts and causally confused policies that ignore environmental information in stochastic environments. This is achieved through two augmentations. First, during training, latent types are not only sampled from the encoder, but also the prior. Because we do not have ground truth trajectories for these prior sampled types, an adversarial loss is used in place of the reconstruction loss. Second, we regularise the mutual information $I(S, \hat{G})$ using a variational information bottleneck (Alemi et al., 2016) to avoid excessive information in $\hat{g}$. RTC combines four losses: the reconstruction loss $\mathcal{L}_{\text{rec}}$, the information bottleneck loss $\mathcal{L}_{\text{ib}}$, the adversarial loss $\mathcal{L}_{\text{adv}}$, and the prior loss $\mathcal{L}_{\text{prior}}$ (see fig. 2):

$$\begin{aligned}
\mathcal{L}_{\text{RTC}} = \ &\mathbb{E}_{\mathcal{D}(\tau)e_\theta(\hat{g}_e|\tau)\pi_\theta(\hat{\tau}_e|\hat{g}_e)}\big[\mathcal{L}_{\text{rec}}(\tau, \hat{\tau}_e) + \beta\mathcal{L}_{\text{ib}} + \lambda_{\text{adv}}\mathcal{L}_{\text{adv}}(\hat{\tau}_e) + \mathcal{L}_{\text{prior}}(\tau)\big] \\
+\ &\mathbb{E}_{\mathcal{D}(\tau)p_{\bar{\theta}}(\hat{g}_p)\pi_\theta(\hat{\tau}_p|\hat{g}_p)}\big[\qquad\qquad\qquad\quad \lambda_{\text{adv}}\mathcal{L}_{\text{adv}}(\hat{\tau}_p) + \mathcal{L}_{\text{prior}}(\tau)\big].
\end{aligned} \tag{4}$$

$p_\theta(\hat{g}_p)$ is a learned prior and $\pi_\theta(\hat{\tau}|\hat{g})$ is shorthand for generating trajectories $\hat{\tau}$ by rolling out the learned control policy $\pi_\theta(\hat{a}|\hat{s}, \hat{g})$ in the environment. Parameters $\bar{\theta}$ are held fixed and $\lambda_{\text{adv}}$ and $\beta$ are scalar weights.

We now introduce the individual terms. First, $\mathcal{L}_{\text{rec}}$ is a reconstruction loss between $\tau$ and $\hat{\tau}_e$, encouraging the hierarchical policy to be distributionally realistic and encode useful information about the trajectory in the inferred type $\hat{g}_e$. The loss $\mathcal{L}_{\text{rec}}$ can take different forms. For example, in section 6.1 we use the BC loss $\mathcal{L}_{\text{rec}}(\tau) = -\log \pi_\theta(a_t|s_t, \hat{g}_e)$ while in section 6.2 we minimise the $L_2$ distance between agent positions in $s_t$ and $\hat{s}_t$. Note that state-based losses like the $L_2$ reconstruction loss require access to a training environment able to resimulate the conditions of the original trajectory $\tau$ as we assume that $\tau$ is still approximately optimal. The loss $\mathcal{L}_{\text{prior}}(\tau) = \mathbb{E}_{\hat{g}_e \sim e_{\bar{\theta}}(\hat{g}_e|\tau)}\left[\log p_\theta(\hat{g}_e)\right]$ optimises the prior to propose *distributionally realistic* types by matching the marginal encoder distribution.

The key algorithmic contribution of RTC is to also optimise the policy under types sampled from the prior (second line in eq. (4)), not only the encoder (first line in eq. (4)). For these prior-sampled types, the reconstruction loss cannot be used as the correct ground truth trajectories are unavailable. Instead, we use the adversarial loss $\mathcal{L}_{\mathrm{adv}}(\hat{\tau}) = \sum_t -\log D_{\boldsymbol{\phi}}(\hat{\boldsymbol{a}}_t, \hat{\boldsymbol{s}}_t)$, where $D_{\boldsymbol{\phi}}(\hat{\boldsymbol{a}}_t, \hat{\boldsymbol{s}}_t)$ is a learned discriminator (see section 2). This reduces conditional type shift because the prior distribution is already used during training. It also reduces causal confusion: Because some types $\hat{\boldsymbol{g}}$ are now sampled independently of the trajectory $\tau$ and hence $\boldsymbol{\xi}$, their information about $\boldsymbol{\xi}$ is now less reliable and the policy is incentivised to rely on $\boldsymbol{s}$ as much as possible.

One can view sampling from the prior as a causal intervention $do(\hat{\boldsymbol{g}})$ in which $\hat{\boldsymbol{g}}$ is changed independently of the environmental factor $\boldsymbol{\xi}$. De Haan et al. (2019) show that causal confusion can be avoided by applying such interventions and optimising the policy to correctly predict the counterfactual expert trajectory distribution, in our case $p_{\mathrm{expert}}(\tau|\boldsymbol{\xi}, do(\hat{\boldsymbol{g}}))$. Unfortunately, we do not have access to this counterfactual trajectory. Instead, we rely on the generalisation of $\boldsymbol{\pi}_{\boldsymbol{\theta}}$ to get us 'close' to such a counterfactual trajectory for types $do(\hat{\boldsymbol{g}})$ and then refine the policy locally using the adversarial objective.

We experimentally found that optimising the policy under prior types is sufficient to improve task performance and avoid causal confusion in hierarchical policies. However, it also eliminated improvements in distributional realism gained through the use of hierarchies, likely because the policy simply learned to ignore the latent type altogether. As solution, we employ a informational bottleneck on types $\hat{\boldsymbol{g}}_e \sim \boldsymbol{e}_{\boldsymbol{\theta}}(\hat{\boldsymbol{g}}_e|\tau)$. This filters information about $\boldsymbol{\xi}$ while still encoding information about $\boldsymbol{g}$ in $\hat{\boldsymbol{g}}_e$, thereby making the information in $\hat{\boldsymbol{g}}_e$ more reliable and useful to the policy. We experimentally show that this, when combined with prior-type sampling during training, achieves improved distributional realism while maintaining excellent task performance. Without prior-type sampling, task-performance degrades considerably, indicating that the bottleneck is insufficient for filtering out information about $\boldsymbol{\xi}$ entirely.

The information bottleneck preferentially filters information about $\boldsymbol{\xi}$, because the control policy $\boldsymbol{\pi}_{\boldsymbol{\theta}}$ also has direct access to it through the visited states $\boldsymbol{s}$. By contrast, information about $\boldsymbol{g}$ can *only* be accessed by the policy through $\hat{\boldsymbol{g}}_e$ and is hence preferably encoded in the bottleneck when information bandwidth costs are applied. We found that both continuous type representations with $\mathcal{L}_{\mathrm{ib}} = \mathrm{KL}[p(\hat{\boldsymbol{g}}_e)\|\mathcal{N}(\hat{\boldsymbol{g}}_e; 0, I)]$ and discrete type representations using straight-through gradient estimation work well in practice (see section 6.2).

To accommodate optimisation under inferred types drawn from both the encoder $\boldsymbol{e}_{\boldsymbol{\theta}}$ and the prior $p_{\boldsymbol{\theta}}$, we split each minibatch $\mathcal{B} = \{\tau^{(b)}\}_b^{N_b}$ of $N_b$ trajectories sampled from $\mathcal{D}$ into two parts. For the fraction $f$ of trajectories in $\mathcal{B}$ the rollouts $\hat{\tau}_e$ are generated from types sampled from the encoder $\hat{\boldsymbol{g}}_e \sim \boldsymbol{e}_{\boldsymbol{\theta}}(\hat{\boldsymbol{g}}_e|\tau)$ and all four losses are optimised (first line in eq. (4)). For the remaining fraction $(1-f)$ of trajectories types are sampled from the prior $p_{\boldsymbol{\theta}}(\hat{\boldsymbol{g}}_p)$ and only $\mathcal{L}_{\mathrm{adv}}$ and $\mathcal{L}_{\mathrm{prior}}$ are optimised (second line in eq. (4)).

Optimisation of $\mathcal{L}_{\mathrm{adv}}$ and $\mathcal{L}_{\mathrm{rec}}$ can either be performed directly, similar to MGAIL (Baram et al., 2016), by using a differentiable environment and reparameterised policies and encoder (Kingma and Welling, 2013) or by treating them as rewards and using RL methods such as TRPO (Schulman et al., 2015; Ho and Ermon, 2016) or PPO (Schulman et al., 2017). The losses $\mathcal{L}_{\mathrm{prior}}$ and $\mathcal{L}_{\mathrm{ib}}$ can always be optimised directly.

## 5 RELATED WORK

Several previous works combine adversarial training with autoencoder architectures in the image domain. Makhzani et al. (2016) use an adversarial loss on the latent variable in place of the KL-regularization used in VAEs. However, this eliminates the information bandwidth regularization for continuous latents which we show to be important for hierarchical imitation learning. Larsen et al. (2016) aim to learn a similarity metric for visual inputs using latent representations of the discriminator. This is valuable for imitation learning from raw images (Rahmatizadeh et al., 2018), but is not required for our experimental domains. Lastly, Chrysos et al. (2018), similar to our work, use an additional autoencoding loss to better capture the data distribution in the latent space. However, they consider denoising images instead of imitation learning under stochasticity.

Hierarchical policies have been extensively studied in RL (e.g., Sutton et al., 1999; Bacon et al., 2017; Vezhnevets et al., 2017; Nachum et al., 2019; Igl et al., 2020) and IL. In RL, they improve exploration, sample efficiency and fast adaptation. By contrast, in IL, hierarchies are used to capture multimodal distributions, improve data efficiency (Krishnan et al., 2017; Le et al., 2018), and enable goal conditioning (Shiarlis et al., 2018). Similar to our work, Wang et al. (2017) and Lynch et al. (2020) learn to encode trajectories into latent types that influence a control policy. Crucially, both only consider deterministic environments and hence avoid the distribution shifts and unwanted information leakage we address. They extend prior work in which the type, or context, is provided in the dataset (Merel et al., 2017), which is also assumed in (Fei et al., 2020). Tamar et al. (2018) use a sampling method to infer latent types. Khandelwal et al. (2020) and Igl et al. (2022) use manually designed encoders specific to road users by expressing future goals as sequences of lane segments. This avoids information leakage but cannot express all characteristics of human drivers, such as persona, and cannot transfer to other tasks. Futures states in deterministic environments (Ding et al., 2019), language (Pashevich et al., 2021), and predefined strategy statistics (Vinyals et al., 2019) have also been used as types.

Information theoretic regularization offers an alternative to learning hierarchical policies using the auto-encoder framework (Li et al., 2017; Hausman et al., 2017). However, these methods are less expressive since their prior distribution cannot be learned and only aim to cluster modes already captured by the agent but not penalize dropping modes in the data. This provides a useful inductive bias but often struggles in complex environments with high diversity, requiring manual feature engineering (Eysenbach et al., 2019; Pathak et al., 2019).

Lastly, TrafficSim (Suo et al., 2021) uses IL to model driving agents and controls all stochasticity in the scene but uses independent prior distributions for separate agents. Hence, while no conditional distribution shift in $p(\hat{\boldsymbol{g}}|\boldsymbol{\xi})$ can occur (as $\boldsymbol{\xi}$ is constant), distribution shifts in $p(\hat{\boldsymbol{g}}^{(i)}|\hat{\boldsymbol{g}}^{(j)})$, and hence the joint marginal $p(\hat{\boldsymbol{g}}^{(1:N)})$ can occur for latent types $\hat{\boldsymbol{g}}^{(i)}, \hat{\boldsymbol{g}}^{(j)}$ of agents $i \neq j$ with $i, j \in \{1 \dots N\}$ and $\hat{\boldsymbol{g}}^{(1:N)} = [\hat{\boldsymbol{g}}^{(1)} \dots \hat{\boldsymbol{g}}^{(N)}]$: when drawn from the encoder, goals $\hat{\boldsymbol{g}}^{(i)}$ and $\hat{\boldsymbol{g}}^{(j)}$ are coordinated through conditioning on the joint agent future, while they are independent when drawn from the prior. They use a biased "common sense" collision avoidance loss, motivated by covariate shift in visited states. Our work suggests that marginal type shift might also explain the benefits gained. In contrast, our adversarial objective is unbiased. See appendix B for more related work on *agent modelling* in multi-agent settings, *behavioural prediction* and *causal confusion*.

## 6 EXPERIMENTS

We show in two stochastic environments with multimodal expert behaviour that i) existing adversarial methods suffer from insufficient distributional realism, ii) existing hierarchical methods cannot achieve good task performance *and* distributional realism and iii) RTC improves distributional realism while maintaining excellent task performance. We discuss differences in *realism*, *coverage* and *distributional realism* in fig. 5.

We compare the following models: MGAIL uses a learned discriminator and backpropagates gradients through the differentiable environment. It also optimises a BC loss as we found this to improve performance. Symphony (Igl et al., 2022), building on MGAIL, utilises future lane segments as manually specified types (see appendix D.3). Our implementation of Symphony outperforms the results from (Igl et al., 2022) due to the additional use of a value function. InfoMGAIL (Li et al., 2017) augments MGAIL to elicit distinct trajectories for different types. This introduces an inductive bias but does not directly penalise mode collapse. Our methods, RTC-C and RTC-D, use a continuous or discrete type respectively. We also perform the ablation Hierarchy-NoPT (*No Prior Training*) which only uses the first line in eq. (4), i.e. $f = 1$. Hierarchy-NoPT is similar to existing hierarchical methods, such as the proprietary TrafficSim (Suo et al., 2021), in that it learns the prior but does not use it during training, only inference. It thereby does not account for distribution shifts in the latent types, as discussed in section 3.

### 6.1 DOUBLE GOAL PROBLEM

In the double goal problem, the expert starts from the origin and creates a multimodal trajectory distribution by randomly choosing and approaching one of two possible, slowly moving goals located on the 2D plane.

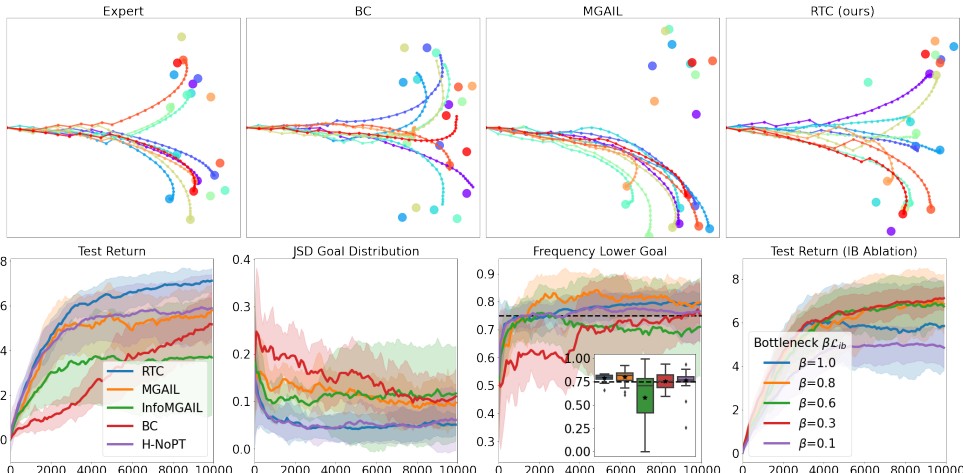

**Figure 3:** *Top:* Visualization of ten randomly sampled goal pairs and associated trajectories. *Bottom:* Training curves, exponentially smoothed and averaged over 20 seeds. Shading shows the standard deviation. We show task performance as 'Test Return' and distributional realism as 'JSD' between the goal distribution of expert and agent (lower is better). 'Frequency Lower Goal' shows the data from which the JSD is computed. The inset shows the distribution at the last training step. Boxes show quartiles, whiskers extreme values, diamonds outliers, and stars the mean.

Stochasticity is introduced through randomized initial goal locations and movement directions. Nevertheless, the *lower* and *upper* goal $\{\boldsymbol{g}_l, \boldsymbol{g}_u\}$ remain identifiable by their location as $y_l < 0$ for $\boldsymbol{g}_l$ and $y_u > 0$ for $\boldsymbol{g}_u$ (see fig. 3). While both goals are equally easy to reach, the expert has a preference $P(G = \boldsymbol{g}_l) = 0.75$. Sufficiently complex expert trajectories prevent `BC` from achieving optimal performance, requiring more advanced approaches. The expert follows a curved path and randomly resamples the selected goal for the first ten steps to avoid a simple decision boundary along the $x$-axis in which experts in the lower half-plane always target goal $\boldsymbol{g}_l$. `RTC` uses the BC loss as reconstruction loss $\mathcal{L}_{\text{rec}}(\tau) = -\log \boldsymbol{\pi_\theta}(\boldsymbol{a}_t|\boldsymbol{s}_t, \hat{\boldsymbol{g}}_e)$ and continuous types. All policies use a bimodal Gaussian mixture model as action distribution. Performance is measured as the number of steps for which the agent is within $\delta = 0.1$ distance of one of the goals. We take $h_s = \text{sign}(y_T)$ of the final agent position $[x_T, y_T]$ to indicate the approached goal and measure distributional realism as the divergence between the empirical distributions, $\text{JSD}\left(p_{\text{agent}}(h_s) \| p_{\text{expert}}(h_s)\right)$. Details can be found in appendix D.1.

Figure 3 shows that `MGAIL` improves task performance compared to `BC`. Our method, `RTC`, improves it further, possibly because given a type, the required action distribution is unimodal. Importantly, `RTC` substantially improves distributional realism, achieving lower JSD values. To analyse this result, we show $p_{\text{agent}}(h_s = -1)$, the frequency of targeting the lower goal. Not only is `RTC`'s average value of $p_{\text{RTC}}(h_s = -1)$ closer to the true value of 0.75, it is also more stable across seeds, resulting in a lower JSD. The bias introduced by `InfoMGAIL` reduces task performance without improving distributional realism. As expected, the ablation `Hierarchy-NoPT` achieves excellent distributional realism through the learned hierarchy but suffers reduced task performance due to unaccounted distribution shifts. Lastly, the rightmost plot of fig. 3 shows that the information bottleneck is necessary.

## 6.2 WAYMO OPEN MOTION DATASET (WOMD)

To evaluate `RTC` on a complex environment we use the *Waymo Open Motion Dataset* (Ettinger et al., 2021) consisting of $487K$ segments of real world driving behaviour. Distributionally realistic agents are critical for driving simulations, for example for estimating safety metrics. Diverse intents and driving styles cause the data to be highly multimodal. Stochasticity is induced through the unpredictable behaviour of other cars, cyclists and pedestrians. We use $\mathcal{L}_{\text{rec}}(\tau, \hat{\tau}) = \sum_t^T \mathcal{L}_{\text{Huber}}(\boldsymbol{s}_t, \hat{\boldsymbol{s}}_t)$ where $\mathcal{L}_{\text{Huber}}$ is the average Huber loss of the four vehicle bounding box corners. More details can be found in appendix D.2.

**Table 1:** Averages and standard deviation over 10 training runs on *WOMD*.

| | Collision rate (%) ↓ | Off-road time (%) ↓ | MinADE (m) ↓ | Curvature JSD ($\times 10^{-3}$) ↓ | Progress JSD ($\times 10^{-3}$) ↓ |
|---|---|---|---|---|---|
| Data Distribution | 1.16 | 0.68 | - | - | - |
| MGAIL | $5.39 \pm 0.68$ | $0.89 \pm 0.12$ | $1.34 \pm 0.08$ | $1.32 \pm 1.48$ | $3.81 \pm 1.29$ |
| Symphony | $6.39 \pm 0.95$ | $0.90 \pm 0.06$ | $1.40 \pm 0.12$ | $0.97 \pm 0.62$ | $6.44 \pm 5.25$ |
| InfoMGAIL - C | $5.21 \pm 0.37$ | $0.89 \pm 0.14$ | $1.29 \pm 0.07$ | $1.24 \pm 0.93$ | $4.40 \pm 1.47$ |
| InfoMGAIL - D | $4.82 \pm 0.29$ | $0.84 \pm 0.10$ | $1.35 \pm 0.11$ | $0.77 \pm 0.44$ | $4.01 \pm 1.45$ |
| Hierarchy-NoPT | $35.08 \pm 0.44$ | $1.83 \pm 0.42$ | $\mathbf{1.12 \pm 0.01}$ | $1.76 \pm 2.05$ | $2.54 \pm 0.63$ |
| RTC - NoIB | $\mathbf{3.88 \pm 0.35}$ | $0.67 \pm 0.04$ | $1.45 \pm 0.14$ | $1.09 \pm 0.62$ | $2.84 \pm 0.65$ |
| RTC - C | $4.23 \pm 0.16$ | $\mathbf{0.68 \pm 0.04}$ | $1.15 \pm 0.10$ | $\mathbf{0.43 \pm 0.06}$ | $\mathbf{2.17 \pm 0.65}$ |
| RTC - D | $4.21 \pm 0.24$ | $0.74 \pm 0.06$ | $\mathbf{1.12 \pm 0.10}$ | $0.89 \pm 0.66$ | $2.56 \pm 0.54$ |

We use the percentage of segments with collisions and time spent off-road as proxy metrics for task performance. Mode coverage is measured by the minimum average displacement error, $\text{minADE} = \mathbb{E}_{\tau \sim \mathcal{D}, \{\hat{\tau}_i\}_i^K \sim \pi_\theta} \left[ \min_{\hat{\tau}_i} \frac{1}{T} \sum_{t=1}^{T} \delta(s_t, \hat{s}_{i,t}) \right]$, where $\delta$ is the Euclidean distance between agent positions and we find the minimum over $K = 16$ rollouts (hierarchical methods use $K$ independently sampled types). Lower *minADE* implies better mode coverage, but does not directly measure the relative frequency of modes, e.g., low probability modes may be overrepresented. To measure distribution matching in driving intent, we use the *Curvature JSD* (Igl et al., 2022): in lane branching regions, such as intersections, it maps trajectories to the nearest lane and extracts its curvature as feature $h_{cur}$. To compute JSD $(p_{\text{agent}}(h_{cur}) \| p_{\text{expert}}(h_{cur}))$, the value of $h_{cur}$ is discretize into 100 equisized bins. To measure the driving style distribution, we extract the progress feature $h_{style} = \delta(\hat{s}_0, \hat{s}_T)$ and use the same discretization to compute the JSD.

Results are provided in table 1. Both versions of `RTC` improve task performance (collisions and off-road events) and distributional realism metrics (minADE and divergences) compared to the flat `MGAIL` baseline and previous hierarchical approaches (`Symphony`, `InfoMGAIL`, `Hierarchy-NoPT`). Both type representations, `RTC-C` and `RTC-D`, perform similarly, showing robustness of `RTC` to different implementations. The advantage of `RTC` in achieving *both* good task performance and distributional realism becomes clearest by comparing it to `Hierarchy-NoPT` and `RTC-NoIB`. While `Hierarchy-NoPT` achieves some improvements in distributional realism, is has nearly an order of magnitude more collisions. This is a consequence of the challenges discussed in section 3, which `RTC` is able to avoid. On the other hand, `RTC-NoIB`, also avoids these challenges and achieves excellent task performance by using prior-sampled types during training. However, as discussed in section 4, it does not improve on distributional realism compared to flat baselines, indicating that the learned policy simply ignores the latent type. Combining prior-type sampling and the information bottleneck achieves better distributional realism and task performance than all baselines.

## 7 Conclusions, Limitations, and Future Work

This paper identified new challenges in learning hierarchical policies from demonstration to capture multimodal trajectory distributions in stochastic environments. We expressed them as *conditional type shifts* and *causal confusion* in the hierarchical policy. We proposed *Robust Type Conditioning* (`RTC`) to eliminate these distribution shifts and showed improved distributional realism while maintaining or improving task performance on two stochastic environments, including the Waymo Open Motion Dataset (Ettinger et al., 2021). Future work will address *conditional* distributional realism by not only matching the marginal distribution $p(\tau)$, but the conditional distribution $p(\tau|\xi)$ under a specific realization of the environment. For example, drivers might change their intent based on the current traffic situation or players might adapt their strategy as the game unfolds. Achieving such conditional distributional realism will also require new models and metrics.

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

# A PROOFS

We restate the theorem and corollary for convenience.

**Theorem 1.** We assume the model $p_{\theta}(\hat{a}|s, a) = \int e_{\theta}(\hat{g}_e|a, s)\pi_{\theta}(\hat{a}|s, \hat{g}_e)d\hat{g}_e$ is achieving optimal reconstruction loss $\mathcal{L}_{\text{rec}} = 0$ on $P_{\mathcal{D}}(s, a)$. The test policy is $p_{\theta}(\hat{a}|s) = \int p_{\theta}(\hat{g}_p)\pi_{\theta}(\hat{a}|s, \hat{g}_p)d\hat{g}_p$ with the marginal encoder $p_{\theta}(\hat{g}) = \mathbb{E}_{P_{\mathcal{D}}}[e_{\theta}(\hat{g}|a, s)]$ as prior distribution. We can say for the training distribution $P(s, a, \hat{g}_e) = P_{\mathcal{D}}(s, a)e_{\theta}(\hat{g}_e|s, a)$ and and testing distribution $P(s, \hat{a}, \hat{g}_p) = P_{\mathcal{D}}(s)p_{\theta}(\hat{g}_p)\pi_{\theta}(\hat{a}|s, \hat{g}_p)$: If $H(\boldsymbol{A}|\hat{\boldsymbol{G}}_e) < I(\boldsymbol{S}, \boldsymbol{A})$ and $H(\boldsymbol{A}, \hat{\boldsymbol{G}}_e) = H(\hat{\boldsymbol{A}}, \hat{\boldsymbol{G}}_p)$, then $I(\boldsymbol{S}, \hat{\boldsymbol{A}}) < I(\boldsymbol{S}, \boldsymbol{A})$.

**Corollary 1.** If $H(\boldsymbol{A}|\hat{\boldsymbol{G}}_e) = 0$, the assumption $H(\boldsymbol{A}, \hat{\boldsymbol{G}}_e) = H(\hat{\boldsymbol{A}}, \hat{\boldsymbol{G}}_p)$ becomes unnecessary in theorem 1 and we have $I(\boldsymbol{S}, \hat{\boldsymbol{A}}) = 0 < I(\boldsymbol{S}, \boldsymbol{A})$.

## A.1 PRELIMINARIES

We denote by $H(X)$ the entropy, by $H(X|Y)$ the conditional entropy, by $I(X, Y)$ the mutual information and by $I(X, Y|Z)$ the conditional mutual information between random variables. Furthermore, the proof is relying on the *interaction information* $I(X, Y, Z)$, an extension of mutual information to three variables. Importantly, the interaction information can be positive or negative. A positive interaction information indicates that one variable explains some of the correlation between the other two while a negative interaction information indicates that one variable enhances their correlation.

Our model $e_{\theta}(\hat{g}_e|a, s)\pi_{\theta}(\hat{a}|s, \hat{g}_e)$ is trained on the dataset $P_{\mathcal{D}}(s, a)$. To achieve minimal reconstruction loss, the model is required to predict $\hat{a} = a$ with certainty, implying $H(\hat{\boldsymbol{A}}|\boldsymbol{S}, \hat{\boldsymbol{G}}_e) = H(\hat{\boldsymbol{A}}|\boldsymbol{S}, \hat{\boldsymbol{G}}_p) = 0$.

At test time, latents are drawn from the prior $p_{\theta}(\hat{g}_p)$ which we assume matches the marginal distribution of the encoder, i.e. $\mathbb{E}_{P_{\mathcal{D}}}[e_{\theta}(\hat{g}_e|a, s)]$, perfectly.

We use the following equalities:

- $I(X, Y, Z) = I(X, Y) - I(X, Y|Z)$ (and permutations as $I(X, Y, Z)$ is symmetric)
- $I(X, Y|Z) = H(X|Z) - H(X|Y, Z)$ (and permutations)
- $H(X|Y, Z) \leq H(X|Y)$
- $H(X, Y) = H(X) + H(Y|X)$ (and permutations)
- $I(X, Y) > 0$

## A.2 PROOF OF THEOREM

During training on the dataset $P_{\mathcal{D}}(s, a)$ the interaction information is positive because $H(\boldsymbol{A}|\hat{\boldsymbol{G}}_e) < I(\boldsymbol{S}, \boldsymbol{A})$:

$$I(\boldsymbol{A}, \hat{\boldsymbol{G}}_e, \boldsymbol{S}) = I(\boldsymbol{S}, \boldsymbol{A}) - I(\boldsymbol{S}, \boldsymbol{A}|\hat{\boldsymbol{G}}_e) = I(\boldsymbol{S}, \boldsymbol{A}) - H(\boldsymbol{A}|\hat{\boldsymbol{G}}_e) + \underbrace{H(\boldsymbol{A}|\boldsymbol{S}, \hat{\boldsymbol{G}}_e)}_{=0} > 0. \tag{5}$$

On the other hand, during testing, we have $I(\hat{\boldsymbol{G}}_p, \boldsymbol{S}) = 0$ because now $\hat{\boldsymbol{G}}_p$ is drawn independently of $\boldsymbol{S}$ from $p_{\theta}(\hat{\boldsymbol{G}}_p)$. Consequently, the interaction information becomes negative:

$$I(\hat{\boldsymbol{A}}, \hat{\boldsymbol{G}}_p, \boldsymbol{S}) = \underbrace{I(\hat{\boldsymbol{G}}_p, \boldsymbol{S})}_{=0} - I(\hat{\boldsymbol{G}}_p, \boldsymbol{S}|\hat{\boldsymbol{A}}) = H(\hat{\boldsymbol{G}}_p|\hat{\boldsymbol{A}}, \boldsymbol{S}) - H(\hat{\boldsymbol{G}}_p|\hat{\boldsymbol{A}}) \leq 0 \tag{6}$$

We also have, similarly to eq. (5),

$$I(\hat{\boldsymbol{A}}, \hat{\boldsymbol{G}}_p, \boldsymbol{S}) = I(\boldsymbol{S}, \hat{\boldsymbol{A}}) - H(\hat{\boldsymbol{A}}|\hat{\boldsymbol{G}}_p)$$

implying

$$I(\boldsymbol{S}, \hat{\boldsymbol{A}}) - H(\hat{\boldsymbol{A}}|\hat{\boldsymbol{G}}_p) \leq 0 < I(\boldsymbol{S}, \boldsymbol{A}) - H(\boldsymbol{A}|\hat{\boldsymbol{G}}_e) \tag{7}$$

and hence, if $H(\boldsymbol{A}|\hat{\boldsymbol{G}}_e) = H(\hat{\boldsymbol{A}}|\hat{\boldsymbol{G}}_p)$, this gives us the desired result

$$I(\boldsymbol{S}, \hat{\boldsymbol{A}}) < I(\boldsymbol{S}, \boldsymbol{A}). \tag{8}$$

By assumption, we have

$$H(\boldsymbol{A}, \hat{\boldsymbol{G}}_e) = H(\hat{\boldsymbol{G}}_e) + H(\boldsymbol{A}|\hat{\boldsymbol{G}}_e) = H(\hat{\boldsymbol{A}}, |\hat{\boldsymbol{G}}_p) = H(\hat{\boldsymbol{G}}_p) + H(\hat{\boldsymbol{A}}|\hat{\boldsymbol{G}}_p). \tag{9}$$

Furthermore, because the marginals of $\hat{\boldsymbol{G}}_e$ and $\hat{\boldsymbol{G}}_p$ are matched, we have $H(\hat{\boldsymbol{G}}_e) = H(\hat{\boldsymbol{G}}_p)$ and hence the required $H(\boldsymbol{A}|\hat{\boldsymbol{G}}_e) = H(\hat{\boldsymbol{A}}|\hat{\boldsymbol{G}}_p)$ for eq. (8) to hold.

Can we remove $H(\boldsymbol{A}, \hat{\boldsymbol{G}}_e) = H(\hat{\boldsymbol{A}}, \hat{\boldsymbol{G}}_p)$ as an assumption? Unfortunately only if $H(\boldsymbol{A}|\hat{\boldsymbol{G}}_e) = 0$, in which case it is automatically true (see next subsection). Otherwise this assumption is needed to make sure that the entropy in the system remains comparable between training and testing.

If $H(\boldsymbol{A}, \hat{\boldsymbol{G}}_e) \neq H(\hat{\boldsymbol{A}}, \hat{\boldsymbol{G}}_p)$, the main result $I(\boldsymbol{S}, \hat{\boldsymbol{A}}) < I(\boldsymbol{S}, \boldsymbol{A})$ could still hold, but one could also construct environments and encoders in which it does not. The reason is that $H(\boldsymbol{A}|\hat{\boldsymbol{G}})$ depends on the distribution $p(\boldsymbol{s}|\hat{\boldsymbol{g}})$ which changes between training, where it is $p(\boldsymbol{s}|\hat{\boldsymbol{g}}_e)$, and testing, where it is $P_{\mathcal{D}}(\boldsymbol{s})$ due to the independent drawing of $\hat{\boldsymbol{g}}_p$. This can be used to construct environments and encoders that change $H(\boldsymbol{A}, \hat{\boldsymbol{G}})$ and $H(\boldsymbol{A}, \hat{\boldsymbol{G}})$ arbitrarily between training and testing, hence making comparing the mutual information $I(\boldsymbol{S}, \hat{\boldsymbol{A}})$ and $I(\boldsymbol{S}, \boldsymbol{A})$ meaningless.

### A.3 PROOF OF COROLLARY

We have $p(\boldsymbol{a}|\hat{\boldsymbol{g}}) = \int_{\boldsymbol{s}} p(\boldsymbol{s}|\hat{\boldsymbol{g}})\pi_{\boldsymbol{\theta}}(\boldsymbol{a}|\boldsymbol{s}, \hat{\boldsymbol{g}})d\boldsymbol{s}$. We also know that $H(\boldsymbol{A}|\boldsymbol{S}, \hat{\boldsymbol{G}}_e) = H(\hat{\boldsymbol{A}}|\boldsymbol{S}, \hat{\boldsymbol{G}}_p) = 0$ and hence $\pi_{\boldsymbol{\theta}}(\hat{\boldsymbol{a}}|\boldsymbol{s}, \hat{\boldsymbol{g}}) \in \{0, 1\}$. Furthermore, if $H(\boldsymbol{A}|\hat{\boldsymbol{G}}_e) = 0$, the action is fully determined by $\hat{\boldsymbol{G}}_e$, i.e. $\pi_{\boldsymbol{\theta}}(\hat{\boldsymbol{a}}|\boldsymbol{s}, \hat{\boldsymbol{g}}) = \pi_{\boldsymbol{\theta}}(\hat{\boldsymbol{a}}|\hat{\boldsymbol{g}}) \in \{0, 1\}$. Hence, because $\int_{\boldsymbol{s}} p(\boldsymbol{s}|\hat{\boldsymbol{g}}_e)d\boldsymbol{s} = \int_{\boldsymbol{s}} P_{\mathcal{D}}(\boldsymbol{s})d\boldsymbol{s} = 1$, the switch from $p(\boldsymbol{s}|\hat{\boldsymbol{g}}_e)$ to $p(\boldsymbol{s}|\hat{\boldsymbol{g}}_p) = P_{\mathcal{D}}(\boldsymbol{s})$ does not impact $p(\boldsymbol{a}|\hat{\boldsymbol{g}})$, so we have $H(\boldsymbol{A}|\hat{\boldsymbol{G}}_e) = H(\hat{\boldsymbol{A}}|\hat{\boldsymbol{G}}_p) = 0$.

The result that $I(\boldsymbol{S}, \hat{\boldsymbol{A}}) = 0$ follows directly from eq. (7) and $I(\boldsymbol{S}, \hat{\boldsymbol{A}}) > 0$.

## B ADDITIONAL RELATED WORK

Unlike our work, *agent modelling* (Grover et al., 2018; Papoudakis and Albrecht, 2020) often assumes knowledge of agent identities in multi-agent systems and aims at learning a useful representation for each identity. In contrast, we neither know the true type $\boldsymbol{g}$ of the imitated agent, nor the identity of external stochastic noise source $\boldsymbol{\xi}$. Furthermore, applications of *opponent modelling* in RL settings (e.g., Papoudakis and Albrecht, 2020; He et al., 2016; Raileanu et al., 2018; Hernandez-Leal et al., 2019; Xie et al., 2020) are generally unconcerned about distributional realism and do not consider distribution shifts.

*Behaviour Prediction* (BP) also forecasts future trajectories. Unless future steps are predicted independently of the evolution of the scene (e.g., not auto-regressively) (Chai et al., 2019; Cui et al., 2019; Phan-Minh et al., 2020; Liu et al., 2021), these methods also suffer from covariate shift in the state visitations (Bengio et al., 2015; Lamb et al., 2016). Furthermore, if hierarchical methods are used to capture the multimodality in the

data (Tang and Salakhutdinov, 2019; Casas et al., 2020; Ivanovic and Pavone, 2019; Salzmann et al., 2020; Yuan et al., 2021; Hu et al., 2021), they are vulnerable to the same marginal and conditional type shifts we consider. While none of these works take these challenges into account, they often use small discrete latent spaces (e.g., Tang and Salakhutdinov, 2019; Ivanovic and Pavone, 2019; Salzmann et al., 2020), mitigating the severity of the distribution shifts and future information leakage by limiting the information bandwidth of latent types. Furthermore, prediction quality metrics such as displacement-based metrics or log-likelihood are less sensitive to yield lower performance due to covariate shift, which primarily impacts interactions with the environment, such as collisions.

As discussed in section 3, the conditional type shift is exacerbated by causally confused policies relying on the latent type for information about environmental noise. Unlike in most literature on causal confusion (De Haan et al., 2019), our nuisance variables are hence not part of the current state, but the learned latent state. Distribution shift is induced not through earlier actions but through sampling from the prior instead of the encoder. Prior work on causal confusion typically relies on problem specific regularization (e.g. Wen et al., 2020; Park et al., 2021) or has access to an expert or task rewards (e.g. De Haan et al., 2019; Ortega et al., 2021). Instead, our work relies on generalisation over latent types to generate counterfactual trajectories. This generalisation is enabled by the information bottleneck and results are refined by the adversarial loss.

## C LIMITATIONS AND SOCIETAL IMPACT

While `RTC` notably improves distributional realism (see section 6), it does not achieve it perfectly, especially in the long tail of the data distribution. This has implications for its use, for example in economic simulations to evaluate policy proposals or in driving simulations to evaluate autonomous vehicles, where this limitation has to be taken into account and the simulation results should not be trusted unconditionally.

As `RTC` is application agnostic, the societal impact depends on where it is used. Here, we focus on agent-based simulations as we anticipate this to create the highest impact. Examples include better policy decisions through economic simulations, safer autonomous vehicles through driving simulations, better AI in games or improved safety precautions for large crowds of people. For other use-cases, e.g., in armed conflicts, the societal impact will depend on the intention of the simulation. Furthermore, we stress once more, that for many use-cases, precautions have to be taken to account for remaining errors in the learned agents.

Lastly, depending on the use case, algorithmic bias has to be taken into account if mode-collapse might be prevented more effectively by `RTC` for certain strata in the population.

## D ADDITIONAL EXPERIMENTAL DETAILS

### D.1 DETAILS ON DOUBLE GOAL PROBLEM

The agent observation

$$\boldsymbol{s}_t = [\boldsymbol{s}_t, \boldsymbol{g}_{l,t}, \boldsymbol{g}_{u,t}, \boldsymbol{a}_{t-1}] \in \mathbb{R}^8 \text{ with } \boldsymbol{s}_t = [x_t, y_t], \ \boldsymbol{g}_{i,t} = [x_{i,t}, y_{i,t}], \ i \in \{u, l\} \tag{10}$$

contains the 2D position of the agent, $\boldsymbol{s}_t$, as well as two marked locations $\boldsymbol{g}_{l,t}, \boldsymbol{g}_{u,t}$ of the lower and upper goal. Because the current agent position cannot uniquely identify the currently selected goal, the observation also contains the last agent action $\boldsymbol{a}_{t-1}$ with the simple transition function $\boldsymbol{s}_{t+1} = \boldsymbol{s}_t + \boldsymbol{a}_t$. The goal locations are randomly sampled at the beginning of each episode. The lower (upper) goal is always located in the lower (upper) half of the $x, y$ plane. Their horizontal and vertical distances from the initial agent position are uniformly sampled within rectangular bounds $x_i^{(g)} \in [1.8, 2.2]$ and $|y_i^{(g)}| \in [0.3, 0.7]$. Each episode has a fixed horizon of $T = 30$ steps over the course of which each goal moves by $\|\boldsymbol{s}_{i,T}^{(g)} - \boldsymbol{s}_{i,0}^{(g)}\| = 0.15$ in a random direction.

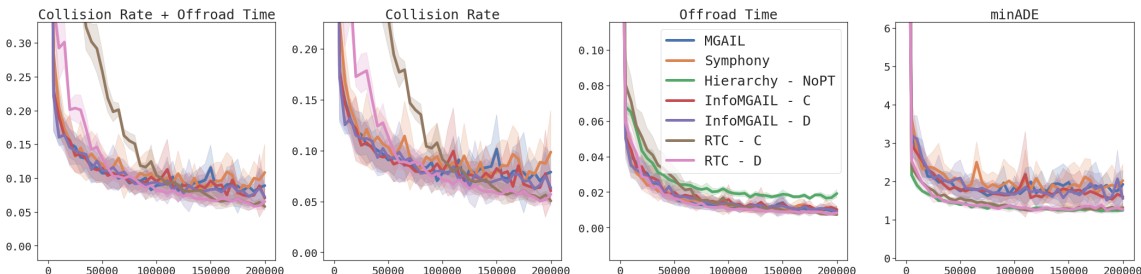

**Figure 4:** *Waymo Open Motion Dataset*: Performance on the validation set during training. Distributional realism metrics are not shown as their evaluation is high variance on the small validation set.

For the first 10 timesteps, the agent randomly resamples the target goal with $P(G = g_l) = 0.75$ to avoid a simple decision boundary along the $x$-axis in which experts in the lower half-plane always target goal $g_l$. The expert action is

$$a_t = 0.1\sqrt{\|\Delta_t\|}\frac{d_t}{\|d_t\|} \quad \text{where} \quad d_t = \begin{bmatrix} 0.1 & 0 \\ 0 & 0.05 \end{bmatrix} \Delta_t \quad \text{and} \quad \Delta_t = (g_t - s_t). \tag{11}$$

The expert approaches the goal faster along the $x$-axis, hence creating a curved path. To avoid over-shooting, the step-size reduces by $\sqrt{\|\Delta_t\|}$ as the agent proceeds towards the goal.

All networks use simple MLPs with two latent layers and a latent dimension of 256. To capture their shape, the discriminator acts on entire trajectories, aggregating across time using max-pooling over a 32 dimensional per-timestep embedding. All policies are parameterised as Gaussian mixture models with two modes. RTC uses a continuous bottleneck of size 2 with additional regularization term $\mathcal{L}_{\text{ib}}^{\beta}(\tau) = \beta KL\left[e_{\boldsymbol{\theta}}(\hat{g}_e|\tau)\|\mathcal{N}(0, I)\right]$ to regulate the information bandwidth of $\hat{g}_e$. We use the BC loss for $\mathcal{L}_{\text{rec}}(\tau, \hat{\tau}_e) = -\log \boldsymbol{\pi_\theta}(\boldsymbol{a}_t|\boldsymbol{s}_t, \hat{\boldsymbol{g}}_e)$.

Batch size is 1024 for training and $10K$ for evaluation. Results shown in fig. 3 are evaluated every 100 steps and exponentially smoothed with a decay rate of 0.9. The learning rate is 0.01 for BC (lower learning rates performed worse) and 0.004 for both MGAIL and RTC, which were tuned independently for values $lr \in [0.02, 0.01, 0.004, 0.002, 0.001]$. $f = 0.5$ was used to split between $\mathcal{B}_{\text{encoder}}$ and $\mathcal{B}_{\text{prior}}$ (no tuning was performed). Lastly, without further tuning, $\lambda_{\text{adv}} = 1$ was used. Training time is about 7h without hardware acceleration.

## D.2 DETAILS ON WAYMO OPEN MOTION DATASET

The Waymo Open Motion Dataset (Ettinger et al., 2021) (published under Apache License 2.0) consists of segments of length $9s$ sampled at $10Hz$. The available training and validation splits in the dataset consist of $487K$ and $49K$ segments each, which are used for training and testing the agent respectively. Due to memory constraints, we filter for segments with less than 256 agents and $10K$ points describing the lane geometry - resulting in $428K$ train and $39K$ test segments. 250 segments from the training split are used for validation to select the training checkpoint for evaluation. In each segment, we learn to control two agents at a frequency of $3.33Hz$, repeating actions three times. Similar to (Igl et al., 2022), the actions of other agents are replayed from the logged data. The collision metric measures the number of segments, in percent, for which at least one pair of bounding boxes overlaps for at least one timestep. The off-road metric similarly detects for how much time the agent's bounding box overlaps with off-road areas.

The state $\boldsymbol{s}_t = [\boldsymbol{s}_t^{(a)}, \boldsymbol{s}_t^{(SS)}, \boldsymbol{s}_t^{(DS)}, \boldsymbol{s}^{(RU)}]$ contains the agent's position and heading $\boldsymbol{s}_t^{(a)}$, static features $\boldsymbol{s}_t^{(SS)}$ such as lane boundaries, expressed as a set of points, dynamical features $\boldsymbol{s}_t^{(DS)}$ such as traffic light states, and the positions and headings of all other road users $\boldsymbol{s}_t^{(RU)}$.

Similarly to section 6.1, we use an additive transition model in which the policy predicts the change in the agents state $\boldsymbol{s}_{t+1}^{(a)} = \boldsymbol{s}_t^{(a)} + \boldsymbol{a}_t$. Dynamic features and roadgraph users are replayed from the logged data, similar to (Igl et al., 2022).

All positions and headings are first normalised to be relative to the observing ego-agent. MLPs are used to encode each object and point individually and per-type max-pooling is used to aggregate over a variable number of inputs. The resulting three embeddings (one for the ego-agent, one for other road-users and one for the scene), each of size 64, are concatenated and passed either to the policy, discriminator or value function, whose encoders are not shared and which consist of MLPs with two latent layers of size 64 for discriminator and value function and 128 for the policy. The inference encoder $\boldsymbol{e}_{\boldsymbol{\theta}}$ for RTC only observes future agent positions $\boldsymbol{s}_{1:T}^{(a)}$ which are each concatenated with an eight-dimensional learned positional embedding and individually encoded to dimension 128 and max-pooled along the time dimension. A Gaussian mixture model with 8 modes was used for all policies, although we find empirically that typically only up to three are used after training.

We train for $200K$ gradient steps and select the model checkpoint for evaluation with the lowest sum of collision rates and off-road time on the validation set. To stabilize training for all methods, we discount gradients through time with $\gamma = 0.9$ and bootstrap from a learned value function every 10 steps. We anneal $f$ from 1 to $f_{\min} = 0.5$ over the course of training. Initially, high values of $f$ encourage meaningful information in $\hat{\boldsymbol{g}}_e$ while lower values address covariate and type shifts and improve performance. A learning rate of 0.0001, which was tuned for MGAIL, was used for all evaluated methods. Each batch contained 24 segments and training was performed on a single V100 (per seed) and required about 4-5 days. We used $\lambda_{\mathrm{adv}} = 4.0$ and $\beta = 0.01$ for $\mathcal{L}_{\mathrm{ib}}(\tau)$ for continuous type representations of size 2. Discrete type representations used three one-hot vectors of size 16, trained using Straight Through gradient estimation. We found performance to be marginally better for three vectors, compared to one, without noticeable performance increases for additional or larger vectors. Smaller vectors with only four values only performed slightly worse. The Huber loss $\mathcal{L}_{\mathrm{rec}}$ uses $\delta = 30$.

### D.3 DETAILS ON SYMPHONY BASELINE

Symphony implements the hierarchical policy proposed in Igl et al. (2022) (called 'MGAIL+H' in their results). Agent types represent high-level driving intent and are expressed as a sequence of road-segments to be followed. They are encoded into a latent vector by expressing them as a fixed-length sequence of points $\{[x_i, y_i]\}_{i=1}^{N_s}$. Each point is concatenated with a positional embedding, then encoded individually, and subsequently max-pooled along the time-dimension. The pooled embedding is provided as additional inputs to both the discriminator and the policy.

During training, lane sequences are extracted from the given data trajectory $\tau$. The prior $p_{\boldsymbol{\theta}}(\hat{\boldsymbol{g}})$, which is used during testing when no ground truth trajectories $\tau$ are available, predicts a categorical distribution over all possible sequences of lane-segments which the agent could follow in a scene. To allow for a variable number of such sequences, the logits are predicted individually per sequence.

### D.4 DETAILS ON INFOGAIL BASELINE

Like RTC, InfoMGAIL (Li et al., 2017) is a general method for learning a hierarchical agent from demonstrations. What makes it a suitable baseline is that, like in RTC, the higher level policy captures a distribution over alternative trajectories that can be taken. It does so in an unsupervised fashion by introducing an

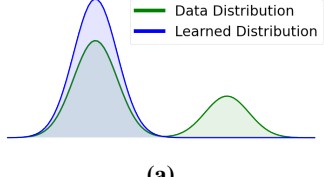 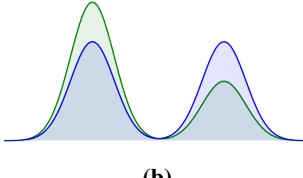 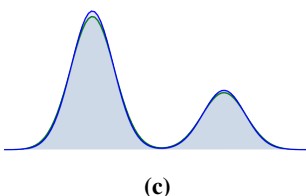

(a)                (b)                (c)

**Figure 5:** Differences between *realism*, *coverage* and *distributional realism*. The data distribution $P(X_D) \in \Delta(\mathcal{X})$ is shown in green, blue denotes a learned distribution $P_\theta(X_L) \in \Delta(\mathcal{X})$. **(a)** Data from learned distribution is *realistic*, i.e. $\mathrm{supp}(X_L) \subseteq \mathrm{supp}(X_D)$, but not *distributionally realistic*. **(b)** The learned distribution achieves *coverage* but not *distributional realism*: the frequencies of modes are not matched. **(c)** The learned distribution is *distributionally realistic*. In practice, the dimensionality of $\mathcal{X}$ is often too high, requiring us to measure distributional realism only in selected features $h(X)$. Consequently, distributional realism in $h(X)$ does not necessarily imply good realism, i.e. task performance.

additional reward that incentivizes the policy to produce state-action pairs from which an additionally trained discriminator (also called 'posterior') can infer the type on which the policy was conditioned. In other words, it rewards the policy for producing distinct trajectories for different types, where the type is drawn from a fixed prior when generating rollouts.

A crucial difference between `InfoMGAIL` and `RTC` is that `InfoMGAIL`'s goal is to disentangle trajectories, but it does not target distributional realism directly. In particular, because the prior from which the types are sampled is fixed, it might not even be able to properly capture the true distribution of trajectories. This is especially true for the uniform discrete prior used in the original `InfoMGAIL` paper, which assumes a uniform distribution over trajectory modes. Furthermore, the additional posterior reward introduces bias, potentially harming task performance. Lastly, because mode collapse is not directly penalized in the additional loss (only 'non-distinctiveness' of trajectories), it might not improve distributional realism at all.

In our experiments, we augment `InfoMGAIL` in several ways:

- We not only try discrete latents, but also continuous ones. For continuous priors we use the same GMM as posterior as we use as prior in RTC.

- We additionally provide the posterior with the initial state as input. Unlike in the examples used in the `InfoMGAIL` paper, we believe that for more complex WOMD data, the current state is insufficient to determine modes.

- To make it comparable in our setup, we optimise it using Info(M)GAIL, i.e. the posterior score of the true type is added as differentiable loss term, not as reward for TRPO. The network architecture of the posterior is the same one as we used for our `MGAIL` discriminator.

- We greatly increase the number of latent dimensions. In (Li et al., 2017), 2 and 3 dimension were used for the two experiments. We tried $d \in [3, 10, 30, 100]$. We also tried $\lambda_1 \in [0.01, 0.03, 0.1, 0.3, 1.0]$ as regularization strength for the additional loss term.

- Lastly, we are also adding a `BC` term to `InfoMGAIL` as we found this stabilizes training greatly.

- In contrast to the original implementation, we are not using pre-training and do not make use of additional shaping rewards.

### D.5 COVERAGE AND DISTRIBUTIONAL REALISM METRICS

While coverage is easy to achieve on the Double Goal Problem, we measure it on the Waymo Open Motion Dataset (section 6.2) using

$$
\mathrm{minADE} = \mathbb{E}_{\tau \sim \mathcal{D}, \{\hat{\tau}_i\}_i^K \sim \boldsymbol{\pi_\theta}} \left[ \min_{\hat{\tau}_i} \frac{1}{T} \sum_{t=1}^{T} \delta(\boldsymbol{s}_t, \hat{\boldsymbol{s}}_{i,t}) \right]
$$

using a fixed number of $K = 16$ rollouts per segment. Intuitively, the more modes are covered by a given agent, the closer one of all $K$ rollouts should be to a given trajectory from the dataset, resulting in a lower *minADE*.

We want to measure distributional realism as the divergence between the expert distribution $p_{\mathrm{expert}}(\tau)$ and the predicted distribution $p_{\mathrm{agent}}(\hat{\tau})$. However, since the space of possible trajectories is far too large to directly measure $\mathrm{JSD}\left(p_{\mathrm{agent}}(\hat{\tau}) \| p_{\mathrm{expert}}(\tau)\right)$, we extract scalar features $h$ from trajectories and measure the divergence on those features. For the Double Goal Problem, we would like to capture whether the agent is approaching $\boldsymbol{g}_l$ or $\boldsymbol{g}_u$, for which we extract $h_s = \mathrm{sign}(y_T)$, i.e. whether the agent is in the lower or upper half of the plane at the last timestep. In the driving domain, we measure progress as the total distance travelled over the $9s$ segment, i.e., $h_{style} = \delta(\hat{\boldsymbol{s}}_0, \hat{\boldsymbol{s}}_T)$. Measuring this distance as a straight line avoids measurement noise through swerving or jittering of the agent. Lastly, to measure high-level intent, i.e. whether the agent prefers going left, right or straight at branching points such as intersections, we follow Igl et al. (2022) and extract as feature $h_{cur}$, i.e., the curvature of the lane segments being followed right after possible branching points in the road.

