# OpenReview forum: "Latent Hierarchical Imitation Learning for Stochastic Environments"
_ICLR.cc/2023/Conference — Submitted to ICLR 2023_

### Official Review · Reviewer_iTsU · 2022-10-23

**Confidence:** 3
**Correctness:** 3
**Technical Novelty And Significance:** 3
**Empirical Novelty And Significance:** 3
**Recommendation:** 8

**Clarity, Quality, Novelty And Reproducibility:**

**Clarity:**
Very clear explanation of the method, motivating examples and limitations/comparisons with other methods, both through text and in the figures.

**Reproducibiliy:**
Method is explaining with high enough detail, but there is no information about hyperparams, resources, training time.

**Novelty:**
The work proposes a new approach to encode variations in policies learned via imitation learning. The core idea builds on top of GAIL and other works encoding agent variability using latent codes, but introduce novelty to separate agent behavior variability vs environment variability.

**Strength And Weaknesses:**

**Strengths:**

- Motivation
	- This is an important and well motivated problem, there are many scenarios in which having agents that cover multiple modes is important, and most environments are stochastic or become stochastic as multiple agents are added.
- Clarity
	- Section 3 gives a clear and intuitive overview of the conditional type shift problem that this work is addressing. I wish some of it was more present in the introduction.
	- Very clear overview of RTC in Figure 3 and generally over setions 3 and 4. I value that authors provide explanations of the effects that every loss is trying to mitigate, and propose strategies for different kinds of environment conditions (such as having vs no having access to environment gradients).
	- Very clear analysis of results, for both benchmarks.
- Novetly
	- The work proposes a new approach to encode variations in policies learned via imitation learning. The core idea builds on top of GAIL and other works encoding agent variability using latent codes, but introduce novelty to separate agent behavior variability vs environment variability.
- Experiments:
	- Approach is tested against strong baselines and sound ablations, with a simple environment that is easily interpretable and a more challenging real-world environment.
	- SOTA results in the Waymo Open Motion Dataset, compared to strong baselines, in terms of performance and mode coverage. The results, particularly the comparison of Hierarchy and RTC shows that the proposed method allows to both cover a high number of modes, but also deal with covariance shift that arises from using a posteriro during training.
- Limitations well addressed in conclusion.

**Weaknesses:**
- More details on claims
	- It would be valuable to provide a reference or more details on some of the claims made in the paper, such as the failure to cover modes of adversarial approaches (in intro).
	- I am not sure I see why encoding encoding all the stochasticity into the agent is a bad approach. In the example in Figure 1, a model-based approach would need to account for stochasticity of the other agents, but the behavior of the agent we are imitating, which only has information of the current and previous states could be controlled by their internal factors, such as risk tolerance to cross or not. A risk tolerant driver will cross when a car is approching, but will stop if the car has already started crossing, avoiding problems. A non-risk tolerant driver will stop directly. I am not sure why we need to explicitly encode the env stochasticity.
	- Isn't it a high assumption that there is a differentiable environment? How well does this method work if having to use reinforcement learning?
- Experiments:
	- For the double-goal environmnet, it would have been interesting to test the effect of environment stochasticity, given that this is one of the claims of the paper. How do the baselines change where there is more/less stochasticity.

**Details**:
- I see that the Lprior, maximizes the likelihood of \hat{g}_{e} and \hat{g}_{p}. Would it make sense to minimize the KL divergence between the 2? Is there a degenerate case where both prior and posterior have high likelihood according to p_theta, but non over-lapping support? What would prevent that?

- Where is the value function in the method proposed here, compared to Symphony? Does it come from the discriminator? Why does it matter if the policy is optimized end-to-end here, instead of using policy gradient?


- Caption in Figure 4 could explain more what the curves in the bottom row mean.

**Summary Of The Paper:**

This paper proposes RTC, an imitation learning approach for stochastic environments that aims to learn policies that avoid mode collapse and cover a wide range of behaviors shown in demonstrations. The paper argues that current methods relying on inferring agent types to explain multi-modal behaviors in stochastic environments suffer from Conditional Type Shift: the agent latents are encoded using a full trajectory and therefore do not account for environment uncertainty, while at test time, agents do not have access to the future, creating a shift between how the agent was encoded at training time and how it is deployed now. To address this, RTC uses proposes to learn a policy by using both the prior and posterior over agent types during training, making sure that trajectories under both distributions are likely according to a discriminator and that the likelihood of both distirbution is high.


**Summary Of The Review:**

The paper addresses an important problem, which is well motivated, and described why it arises. While there are some details (see weakness/details) that I would really like clarified, the approach for RTC addresses some of the limitations of existing IL approaches with latent codes for agent type. The results, both the simple environment to provide more intuition about the method and the Waymo Open Motion show the value of this method. Hence, I think this paper should be accepted.

---

> ### Author Response · Authors · 2022-11-18
> **Thank you!**
>
> Thank you for your positive and helpful review!
>
> > Section 3 gives a clear and intuitive overview of the conditional type shift problem that this work is addressing.
>
> Thank you very much!
> We updated section 3 to include a more rigorous justification of our claims (as asked for by reviewer MSid) and a more precise educational example.
> We hope the new version improves the clarity further.
>
> > It would be valuable to provide a reference or more details on some of the claims made in the paper, such as the failure to cover modes of adversarial approaches (in intro).
>
> We added some references w.r.t mode collapse in adversarial methods, both in IL and more generally.
>
> > I am not sure I see why encoding encoding all the stochasticity into the agent is a bad approach. In the example in Figure 1, a model-based approach would need to account for stochasticity of the other agents, but the behavior of the agent we are imitating, which only has information of the current and previous states could be controlled by their internal factors, such as risk tolerance to cross or not.
> A risk tolerant driver will cross when a car is approching, but will stop if the car has already started crossing, avoiding problems. A non-risk tolerant driver will stop directly. I am not sure why we need to explicitly encode the env stochasticity.
>
> To clarify, we want to _avoid_ encoding the env stochasticity in the latent type. What remains, i.e. what ideally is encoded in the type, is only things like risk tolerance or other internal agent features.
>
> The problem is that when the type also encodes a specific realisation of the env stochasticity (e.g. "the other car is approaching very slowly") the agent might learn to rely on the latent type for this information instead of taking this information from the current state.
> This is fine during training when the encoded information matches what happens in the data.
> However, at test time, this information is randomly drawn and hence might not match.
>
> For example, the randomly drawn prior type might encode "other car is approaching slowly" while in reality the other car is approaching quickly.
> If the agent relies on the latent type, it might enter the intersection and collide.
>
> Not sure if this answer the question or if we misunderstood your comment.
>
> Please also see the new example in section 3 which might help make things more precise.
>
> > Isn't it a high assumption that there is a differentiable environment? How well does this method work if having to use reinforcement learning?
>
> In principle the method should be perfectly compatible with RL.
> For example [1] also combines variational and adversarial methods and use TRPO for optimisation.
> In terms of algorithm, the key distinction to RTC is that RTC also uses simultaneous optimisation under prior sampled types, but this should not impact whether it can be optimised by RL.
>
> Our choice to use a differentiable environment was purely motivated by the Symphony baseline on the WOMD dataset which uses a differentiable environment.
> One issue with WOMD is that the data is quite large, allowing only (relatively) small batch sizes on GPUs. So not having to use RL (which has a higher gradient variance) helps stabilise training.
>
> [1] Wang, Ziyu, et al. "Robust imitation of diverse behaviors." Advances in Neural Information Processing Systems 30 (2017).
>
>
> > For the double-goal environmnet, it would have been interesting to test the effect of environment stochasticity, given that this is one of the claims of the paper. How do the baselines change where there is more/less stochasticity.
>
> We agree, this would be a nice ablation and we will include this in a future version of the paper.
> We do, however, believe our experiments sufficiently show the failure case in stochastic environments and how RTC is able to overcome it.

---

> > ### Author Response · Authors · 2022-11-18
> > **Continuation of reply.**
> >
> > > I see that the Lprior, maximizes the likelihood of \hat{g}{e} and \hat{g}{p}. Would it make sense to minimize the KL divergence between the 2? Is there a degenerate case where both prior and posterior have high likelihood according to p_theta, but non over-lapping support? What would prevent that?
> >
> > Note that L_prior only trains the prior (i.e. $\hat{g}_p$) and the encoder (i.e. $\hat{g}_e$) is regularised using L_ib.
> >
> > But in theory yes, you are right, it would make sense to minimize the KL between the two.
> > However, because we use a GMM for the prior, this KL could not be computed analytically. The split between L_ib and L_prior allows us to apply the KL with a N(0,1) Gaussian analystically to the encoder (as L_ib) and train the prior using samples (using L_prior).
> >
> > The degenerate case shouldn't be possible:
> > 1. The prior is trained to be mode-covering, so its support should be larger than the one of the encoder as it is trained to match the marginal distribution of the encoder.
> > 2. By using a flexible GMM model, our prior is quite expressive and should be able to fit well, even to a multi-modal marginal distribution $\mathbb{E}_{g, \xi}\left[e_\theta(\hat{g}_e|\tau)\right]$.
> >
> > > Where is the value function in the method proposed here, compared to Symphony? Does it come from the discriminator? Why does it matter if the policy is optimized end-to-end here, instead of using policy gradient?
> >
> > Yes, it comes from the discriminator and predicts the expected future discounted discriminator scores.
> > End-to-end optimisation has a lower gradient variance than the RL loss, helping with smaller batch sizes: the WOMD data is quite big.
> > That said, we did not try RL based optimisation, but simply followed the baseline on WOMD, namely Symphony [1].
> >
> > [1] Igl, Maximilian, et al. "Symphony: Learning Realistic and Diverse Agents for Autonomous Driving Simulation." arXiv preprint arXiv:2205.03195 (2022).
> >
> > > Caption in Figure 4 could explain more what the curves in the bottom row mean.
> >
> > Thank you, we added some additional information.
> >
> > > Reproducibiliy: Method is explaining with high enough detail, but there is no information about hyperparams, resources, training time.
> >
> > Apologies, we added these to the appendix but only submitted the appendix in the supplementary material.
> > The updated version of the paper now also includes the appendix in the main pdf.

---

### Official Review · Reviewer_JuGQ · 2022-10-27

**Confidence:** 3
**Correctness:** 3
**Technical Novelty And Significance:** 3
**Empirical Novelty And Significance:** 3
**Recommendation:** 5

**Clarity, Quality, Novelty And Reproducibility:**

Clarity: Fair
Quality: Good
Novelty: Good
Reproducibility: Not sure

**Strength And Weaknesses:**

Strength:
1. The author considered a very comprehensive treatment of distribution shift, while most previous work focuses on covariate shift only.

Weaknesses:
2. I think the method section (section 4: Robust Type Conditioning) is too short, the different components of the loss are not fully explained, which only last for a bit more than 1 page.  For instance

2.1 Why the reconstruction loss prevents mode collapse by penalizing the agent for being unable to mimic \pi?
2.2 Why the adversarial loss could eliminate marginal and conditional shift?
2.3 Could you explain more about the information bandwidth of \hat{g}_e?  Why the prior on agent type could filter out the environment (external) noise?

**Summary Of The Paper:**

Mode collapse is a problem in imitating multi-modal trajectories. Hierarchical method use latent code to represent the agent type or context variable which is inferred during the training and later used at test time, and the policy should be conditioned on the latent code. Context variable include internal (intrinsic setting of agents) and external factors (coming from environment)

Beyond Covariate shift, the author considers marginal type shift where the prior distribution of agent type changes, where conditional type shift is the external context variable is dependent on the internal context variable. Adversarial imitation learning optimize the policy via a discriminator to distinguish between true trajectory and learned trajectory in a min max fashion where the policy try to imitate the observed trajectory and the discriminator try to distinguish between them. By dividing the data input to be the policy sampled trajectory and the real observed trajectory, the discriminator classification loss could be used as reward for a reinforcement learning agent.

The author proposed Robust Type Conditioning by incorporating prior during training, which is summarized into a 4-term loss function evaluated on prior policy and encoded policy.


Results are tested on Waymo Open Motion dataset.



**Summary Of The Review:**

Adding latent variables to imitation learning is crucial for mimicking multimodal behaviors, as well as being able to generalize better in terms of distribution shift.

The author offers a very comprehensive view about tackling the problem with fair amounts of experimental efforts, however, I think it is not very well explained, how each of the individual aspects were tackled, for instance, if one remove one of the loss terms, how much the performance will deteriorate?
What is the theoretical contribution of this paper? Is that learning the prior of agent type?

---

> ### Author Response · Authors · 2022-11-18
> **Thank you!**
>
> Thank you for your feedback and review, helping us to improve the paper!
>
> Based on your, and the other reviewers' feedback, we updated the paper, in particular section 3 and 4, to improve clarity and theoretical rigour.
>
> Below we will address your individual points:
>
> > I think the method section (section 4: Robust Type Conditioning) is too short.
>
> We agree that more space would allow us to expand further on the explanations, but we are unsure where it would be best to reduce content. in particular, section 3 is critical to our contributions.
>
> > 2.1 Why the reconstruction loss prevents mode collapse by penalizing the agent for being unable to mimic \pi?
>
> We updated section 4 to be clearer the roles of the different losses.
> To answer your question: If the agent is able to mimic a given trajectory $\tau$, this means that there is some latent type which will generate such a trajectory.
> Consequently, during testing, whenever such a latent type will be drawn, the corresponding mode that $\tau$ belongs to will be covered.
>
> If, one the other hand, $\tau$ cannot be reconstructed (and has hence high reconstruction loss), then the agent cannot reproduce the corresponding mode.
>
> Consequently, because mode collapse leads to a high reconstruction loss, this penalises it.
>
>
> > 2.2 Why the adversarial loss could eliminate marginal and conditional shift?
>
> We removed the marginal type shift from the paper to focus on the conditional type shift.
> We also updated section 4 and removed the misleading sentence.
> The adverarial loss by itself cannot eliminate type shift, but allows training under the prior distribution. Because the prior distribution is used at test time, already using it during training removes and shift in distribution between training and testing.
>
> > 2.3 Could you explain more about the information bandwidth of \hat{g}_e?
>
> Using the variational information bottleneck [1] for continuous latent types leads to a higher loss L_ib the more information is encoded in the latent $\hat{g}_e$.
> On the other hand, using a discrete 1-hot latent type enforces a hard upper bound on the amount of information that can be encoded in $\hat{g}_e$.
>
>
> [1] Alemi, Alexander A., et al. "Deep variational information bottleneck." arXiv preprint arXiv:1612.00410 (2016).
>
> > Why the prior on agent type could filter out the environment (external) noise?
>
> We updated section 4 to improve clarity on this. To quote from the new version:
>
> "This [training under prior samples] reduces conditional type shift because the prior distribution is already used during training. It also reduces causal confusion: Because some types $\hat{g}$ are now sampled independently of the trajectory $\tau$ and hence $\xi$, their information about $\xi$ now less reliable and the policy is incentivised to rely on $s$ as much as possible."
>
> I.e. training under the prior does not by itself filter out external noise, but incentivices the policy to ignore any such information in the type as it is unreliable.

---

> > ### Comment · Reviewer_JuGQ · 2022-12-09
> > **reply to question 2.1 makes sense**
> >
> > Thanks to the author for making this clear, the explanation should have been in the submitted draft.

---

### Official Review · Reviewer_MSid · 2022-11-01

**Confidence:** 5
**Correctness:** 2
**Technical Novelty And Significance:** 2
**Empirical Novelty And Significance:** 1
**Recommendation:** 3

**Clarity, Quality, Novelty And Reproducibility:**

- The quality of the paper is fair. The paper lacks rigorous treatment of the previous works and makes unsubstantiated claims about the problem in those works without providing adequate evidence. It then uses these problems as motivation to propose the new approach which appears more adhoc than inspired from these problems. The lack of adequate empirical support for the claims of  existence of conditional and marginal shifts and lack of comparisons with strong baselines further decreases the quality of this work.
- The paper is easy to follow for most parts but there are several statements or lack thereof that decreases the clarity of paper.
    - For instance, in the introduction, It is not clear what is meant by inferring types from future trajectories. Does it signify the use of entire trajectory for inferring the latent vector (so the future mean all transitions of trajectory known in advance)?
    - Next, access to a differentiable environment is strong requirement and it is not clear if it only applies to the compared MGAIL baseline or also used for the proposed approach.
    - In eq 2 why does $\mathcal{L}_{prior}(\tau)$ appear twice?
    - Presentation of plots is very complicated e.g. frequency lower goal in Figure 4 also has box plots belonging to different methods which are hard to read.
    - Why is it required to have last action as part of state for double goal problem but not WOMD?
    - Why is adversarial training of only sampled prior (i.e. no training based on posterior sample) not enough?
- Given that most terms (Except the prior based adversarial training) have been explored and studied previously, the overall originality of the work is very limited. This limitation is further exacerbated by the lack of support for the claims of existence of conditional and marginal shifts.

**Strength And Weaknesses:**

Strengths:
-----------
+ Imitation learning in stochastic environments is an important problem as most real-world environments have stochastic elements and an approach that can disentangle stochasticity in environment from stochasticity in agent behavior is very useful direction.
+ The proposed method show consistent improvement in performance across different metrics over considered baselines.
+ Effective evaluation of imitation learning method is an open issue and the considered metrics, especially Average displacement error and JSD based metrics are interesting and insightful to use for evaluation.

Weaknesses:
---------------
- The proposed approach heavily relies on the insight that existing type conditioned approach leads to causally confused agents as the design of such methods require the agents to infer the stochasticity of environment from the latent type encodings whose primary purpose is to provide agent-specific information. However, this entire premise is not given a rigorous treatment and presented very loosely with anecdotal examples without any theoretical or empirical  grounding. This makes the identification of conditional distribution shift as a reinterpretation of previous works in light of the proposed approach rather than a concrete motivation. For instance, the example of pass/yield is used to claim that the policy stops using current observation to infer stochasticity and only relies on latent type. Without establishing theoretically or empirically, that existing strong baselines indeed end up doing this, such a claim is not substantiated.
- The main contribution of this work is the objective function Eq 2. that is proposed to mitigate the marginal and the conditional distribution shifts. The reconstruction and adversarial loss have been extensively used by previous imitation learning works [1] even for hierarchical type-conditioned policies [2]. Effectiveness of Information bottleneck term has also been studied previously in the context of imitation learning [3]. Further, the explanation of why this term, while helpful in gaining most of empirical performance (see later), contributes to the disentanglement of $\xi$ and type is unclear and adhoc. The use of sampled prior for adversarial term is novel but it is highly incremental contribution given the existence of previous works on type-conditioned imitation learning. Again, it is not clear why introducing this term in effect, mitigates the marginal shift.
- There are several strong works that are either missing citations [4.5,6] or discussed in a very hand-wavy manner without providing comparisons with them [2,6]. Further, a great weakness of the paper stems from the unsubstantiated claims by the authors claim that hierarchical methods such as [2,6,7] give rise to conditional and marginal shifts . It is imperative that such statements be backed up with rigorous analysis of those methods to effectively showcase the existence of identified issues.
- Due to the above limitations, the burden of the work lies on empirical support, however, the paper has several shortcomings in this area:
    - Lack of comparisons with strong baselines [2,4,5,6,7] is a big miss. It is imperative to show the efficacy of approach as compared to these works as they can be considered as different variants of this approach.
    - The authors mention that an ablation Hierarchy-NoPT which considers no prior training is similar to existing hierarchical methods but it does not cover the methods cited here. Further, if one considers Hierarchy-NoPT as representative of existing methods, then the improvement with prior training is marginal and hence full approach has very limited contribution.
    - Minor follow-up: Given the high performance on  minADE for Hierarchy-NoPT in WOMD dataset, the high collision rate seems like a bug and the reasoning for this is not adequate and seems adhoc.
    - On the double -goal problem, the hyper-parameter for IB is shown to be very sensitive which is a limiting factor as it may not be easy to tune this hyper-parameter. Further, given that this term affects the performance more than any other objective and it is very sensitive, comparisons of the form baselines + IB becomes very useful. For instance, will [2] + IB provide superior performance than proposed approach?
    - Ablation is only provided on toy task however, there is a wide gap in the characteristics of double goal problem and WOMD and hence an ablation on WOMD is also required. Plus, further granular ablation is needed for ex. does reconstruction + IB provide as good performance?
    - It is mentioned that learning rate was only tuned on MGAIL to 0.0001 and then used for all methods. This is a strange design choice and tuning should be done specific to each method.
    - While there is performance improvement across different metrics, it is very difficult to establish that this gain has anything to do with the mitigation of conditional and marginal distribution shifts induced by other methods, which fails to support the main claims of the paper.

[1] Imitation Learning: Progress, Taxonomies and Opportunities, Zheng et. al. 2021.

[2] Robust Imitation of Diverse Behaviors,  Wang et. al. Neurips 2017.

[3] Variational Discriminator Bottleneck: Improving Imitation Learning, Inverse RL, and GANs by Constraining Information Flow, Peng et. al. ICLR 2019.

[4] Imitation Learning from Visual Data with multiple intentions, Tamar et. al. ICLR 2018

[5] Triple-GAIL A multi modal imitation learning framework with Generative Adversarial Nets, Fei et. al. 2020

[6] Multi-modal Imitation Learning from Unstructured Demonstrations using Generative Adversarial Nets, Hausman et. al. Neurips 2017

[7] Learning a Multi-modal policy via Imitating Demonstrations with Mixed Behaviors, Hsiao et. al. Neurips Workshop 2018

**Summary Of The Paper:**

The paper focuses on the problem of imitation learning in stochastic environments. The stochasticity in the environment is characterized by the parameter  $\xi$, that is independent of the expert/agent.
The goal of this work is to build imitation agent that achieve distributional realism — match the expert distribution covering all modes of demonstrations while achieving good task performance.
As a solution approach, the authors attempt to build on and improve previous works on designing hierarchical policies - conditioning the imitation policy on type of expert demonstration- to achieve the stated goal.
The hypothesis is that the demonstrations exhibit multiple modes due to difference in types of expert (where type models expert-specific features e.g. persona, goal, strategy or belief) and such types may be sampled from demonstration of available as labels in the data. The authors argue that existing works in type-conditioned imitation learning (especially when such types are latent and sampled from data) either fail to achieve satisfactory mode coverage or compromise task performance. The authors identify two types of distribution shift in the inferred types - marginal (mismatch between posterior and prior type samples)  and conditional (mismatch arising due to the inability of existing type encoders  to disentangle environment noise and agent specific features) - as the reason for inducing causal confusion in the agent, thereby degrading distribution matching and task performance. The authors propose new objective that aims to mitigates these shifts by allowing to train with types sampled from both posterior and prior (unlike previous methods which only consider posterior types in training) and augmenting the imitation loss with information bottleneck term to address the conditional shift stated above. Empirical investigation on a toy double goal problem and more complex Waymo Open Motion dataset is presented to discern various components of the proposed objective and provide comparisons with representing adversarial IL baselines in terms of task performance and distribution matching.

**Summary Of The Review:**

While the paper deals with an important problem of imitation learning in stochastic environments, the proposed approach is highly incremental and adhoc. The identification of marginal and conditional distributional shifts appears as a reinterpretation rather than motivations grounded in theoretical or empirical evidence. This coupled with multiple major shortcomings in the empirical exercise makes this contribution below par in its current form and informs my current assessment.

---

> ### Author Response · Authors · 2022-11-18
> **Thank you!**
>
> We thank the reviewer for their helpful and detailed feedback.
>
> ## Highlights
>
> ### Theoretical treatment of conditional type shift
>
> We agree with your point that section 3 (the discussion of the conditional type shift) was not rigorous enough and have therefore rewritten it to provide a much more theoretical argument as well as a concrete and precise example for the failure case tackled in this paper.
> We also modified section 4 to improve clarity.
>
> We kindly ask you to have another look and would greatly appreciate any feedback on the new version.
>
> ### Experimental evidence
>
> We believe you misunderstood the experimental evidence and would like to clarify this as it significantly impacts the size of improvement achieved by our method.
> We also hope the new (and hopefully clearer) section 3 can prevent such a misunderstandings in the future.
>
> The high collision rate of Hierarchy-NoPT is not a bug, but the *main motivation of the paper* as it is caused by the conditional type shift and causally confused policy described in section 3 and tackled by RTC.
> Contrary to your argumentation, a high collision rate *is* compatible with low minADE: To achieve low minADE, only *one of 16* trajectories needs to be close (in euclidian space) to the ground truth trajectory and most or even all 16 trajectories could contain collisions without increasing the minADE.
>
> Furthermore, the quantitative results from both experiments agree:
> * Flat methods such as MGAIL achieve good task performance.
> * “Naive” hierarchical methods such as H-NoPT (which does include the bottleneck!) achieve good distributional realism but bad task-performance.
> * Only RTC, which combines the IB with training under prior sampled types, achieves both good task performance and distributional realism.
>
> ## Individual comments
>
> Below, we address your individual points.
>
> > The proposed approach heavily relies on the insight that existing type conditioned approach leads to causally confused agents [...]. However, this entire premise is not given a rigorous treatmen [...].
>
> As discussed, we agree and updated section 3 to include a more rigorous treatment.
>
> > The main contribution of this work is the objective function Eq 2. [...].
>
> Our two main contributions are:
> 1. The (now much more formal) definition of the problem that arises for IL under stochastic environments (see section 3)
> 2. Proposing RTC so solve this problem. We agree that the individual building blocks (adversarial loss, reconstruction loss and information bottleneck) are not novel by itself. What we see as the main novelty is the prior-sampled policy training and the appropriate combination with the other building blocks to avoid the problems described in section 3.
>
> > Further, the explanation of why this term, while helpful in gaining most of empirical performance (see later), contributes to the disentanglement of  and type is unclear and adhoc.
>
> We updated section 4 to improve the explanation.
>
> Importantly (as discussed above), we **disagree that the IB gains most of the empirical performance**: Only using the IB corresponds to "Hierarchy-NoPT", which performs extremely poorly.
>
> > The use of sampled prior for adversarial term is novel but it is highly incremental contribution given the existence of previous works on type-conditioned imitation learning.
>
> We agree that the architectural change is small compared to prior works, but disagree that this makes it necessarily incremental as it significantly improves performance and is (hopefully now) well motivated through the updated sections 3 and 4.
>
> > Again, it is not clear why introducing this term in effect, mitigates the marginal shift.
>
> We agree. The marginal type shift is an orthogonal problem and we removed it from the paper to focus on the conditional type shift.

---

> > ### Author Response · Authors · 2022-11-18
> > **Continuation of reply.**
> >
> > > There are several strong works that are either missing citations [4.5,6] or discussed in a very hand-wavy manner without providing comparisons with them [2,6]. Further, a great weakness of the paper stems from the unsubstantiated claims by the authors claim that hierarchical methods such as [2,6,7] give rise to conditional and marginal shifts. It is imperative that such statements be backed up with rigorous analysis of those methods to effectively showcase the existence of identified issues.
> >
> > Thank you for pointing out [4,5] we included them in the related work.
> > [6] was already included.
> >
> > Work [6] appears to be quite similar to InfoGAIL, which we do compare against in the form of InfoMGAIL: Both InfoMGAIL and [6] use an information-theoretic term to incentivise distinguishability of generated trajectories.
> >
> > We believe both [2] and [7] are similar enough to "Hierarchy-NoPT" to make the qualitative statement that they fail in stochastic environments (see the high collision rate of Hierarchy-noPT).
> > All three algorithms train a VAE encoder in addition to the policy and condition the policy on the encoded trajectories.
> > Note that "Hierarchy-NoPT" uses the L_ib which is equivalent to the KL loss in VAEs!
> >
> > [2] also trains a state decoder, but this seems unrelated to the issues discussed here -- if anything, it should make the discussed problems worse as it forces the encoder to capture environmental noise (exactly the information we don't want in the latent state)
> >
> > Crucially, none of these methods are evaluated in stochastic environments and none of these methods use prior-sampled types, which we show to be crucial in stochastic environments.
> >
> > We hope the new section 3 with the example also helps here because it makes the problem clearer.
> >
> > [2] Robust Imitation of Diverse Behaviors, Wang et. al. Neurips 2017.
> >
> > [6] Multi-modal Imitation Learning from Unstructured Demonstrations using Generative Adversarial Nets, Hausman et. al. Neurips 2017
> >
> > [7] Learning a Multi-modal policy via Imitating Demonstrations with Mixed Behaviors, Hsiao et. al. Neurips Workshop 2018
> >
> >
> > > Lack of comparisons with strong baselines [2,4,5,6,7] is a big miss. It is imperative to show the efficacy of approach as compared to these works as they can be considered as different variants of this approach.
> >
> > We believe we show in our experiments that prior-conditioned training is crucial for good task performance of hierarchical methods in stochastic environments. None of these previous methods make use of this, so we disagree that RTC can be seen as only a variant of these previous methods.
> >
> > > The authors mention that an ablation Hierarchy-NoPT which considers no prior training is similar to existing hierarchical methods but it does not cover the methods cited here.
> >
> > We disagree (see discussion above). Could you please clarify which feature of the listed methods is not covered so we can include a representative method in our experiments?
> >
> > > Further, if one considers Hierarchy-NoPT as representative of existing methods, then the improvement with prior training is marginal and hence full approach has very limited contribution.
> >
> > We disagree (see discussion in the beginning of regarding the experimental results).
> >
> > > On the double-goal problem, the hyper-parameter for IB is shown to be very sensitive which is a limiting factor as it may not be easy to tune this hyper-parameter.
> >
> > We believe our figure was misleading w.r.t the sensitivity as it was varying $\beta$ over several orders of magnitude. We updated the figure to show that performance is stable over approximately half an order of magnitude (from 0.3 to 0.8), which we believe is sufficiently stable to not cause problems.
> >
> > > Further, given that this term affects the performance more than any other objective and it is very sensitive, comparisons of the form baselines + IB becomes very useful. For instance, will [2] + IB provide superior performance than proposed approach?
> >
> > We disagree that it affects the performance more than any other objective or that is sensitive (see clarifications above).
> > Furthermore, we believe [2] already includes the IB loss as it the same as $D_{KL}[q_\phi(z|x^i_{1:T_i}\|p(z)]$: A KL divergence with a multivariate Gaussian distribution.
> >
> > > Ablation is only provided on toy task however, there is a wide gap in the characteristics of double goal problem and WOMD and hence an ablation on WOMD is also required. Plus, further granular ablation is needed for ex. does reconstruction + IB provide as good performance?
> >
> > We also added the ablation study `RTC-NoIB` on WOMD.

---

> > > ### Author Response · Authors · 2022-11-18
> > > **Continuation of continuation of reply.**
> > >
> > > > It is mentioned that learning rate was only tuned on MGAIL to 0.0001 and then used for all methods. This is a strange design choice and tuning should be done specific to each method.
> > >
> > > Initial experiments indicated that all methods were robust to reasonable changes in this hyperparameter, so we fixed it early due to computational constraints. Furthermore, our decision should benefit the baselines more than RTC, hence we believe our experiments provide a fair comparison.
> > >
> > > > While there is performance improvement across different metrics, it is very difficult to establish that this gain has anything to do with the mitigation of conditional and marginal distribution shifts induced by other methods, which fails to support the main claims of the paper.
> > >
> > > We hope the updated sections 3 and 4 make this connection clearer.
> > > In particular, without conditional type shift, it could not be explained why training under the prior distribution would improve performance by such a large amount (when comparing Hierarchy-NoPT with RTC).
> > >
> > > > For instance, in the introduction, It is not clear what is meant by inferring types from future trajectories. Does it signify the use of entire trajectory for inferring the latent vector (so the future mean all transitions of trajectory known in advance)?
> > >
> > > Yes, during training this is possible as we have access to trajectories in the dataset. During testing we do not require future trajectories as we draw types from the prior.
> > >
> > > > Next, access to a differentiable environment is strong requirement and it is not clear if it only applies to the compared MGAIL baseline or also used for the proposed approach.
> > >
> > > It is used for all approaches in order to change as little as possible between methods to provide a fair comparison.
> > > In this, we are following the Symphony paper [1].
> > >
> > > [1] Igl, Maximilian, et al. "Symphony: Learning Realistic and Diverse Agents for Autonomous Driving Simulation." arXiv preprint arXiv:2205.03195 (2022).
> > >
> > > > In eq 2 why does L_prior appear twice?
> > >
> > > To indicate that it is trained on the full batch of trajectories, not only those sampled in the first line. (Note that the loss L_prior contains the expectation over $\hat{g}_e$ in its definition!)
> > >
> > > > Why is it required to have last action as part of state for double goal problem but not WOMD?
> > >
> > > It is needed because double-goal allows the expert to repeatedly change the target goal, effectively making it a POMDP for the imitation agent.
> > > Providing the last action provides the necessary information to turn it into an MDP.
> > > We don't believe this problem exists or is relevant on WOMD.
> > >
> > > > Why is adversarial training of only sampled prior (i.e. no training based on posterior sample) not enough?
> > >
> > > In this case the type cannot provide any useful information to the policy and would be ignored (since it would always be drawn completely random). One could counteract this by encouraging the policy to generate different outputs for different prior types, which is what e.g. InfoMGAIL does (which we comapare against).

---

> ### Comment · Reviewer_MSid · 2022-12-11
> **Too many changes; weak contribution and presentation of revised version**
>
> I thank the authors for their efforts towards addressing my comments in detail and revising the paper. After carefully reading the revised version of the paper, I still do not think that the contributions in this paper are technically significant and/or rigorously treated.
> - As per reviewer's own admission, marginal type shift was incorrect to be included. Further, the motivation and treatment of conditional type shift is still hand wavy and based on changes made by authors, clearly acknowledged as being an after thought.
> - While the authors have addressed some empirical questions, I am still not convinced by the performance gain. Specifically, the role of IB and hyper-parameter sensitivity are major issues and the presentation of proposed technique, in its current form, lacks evidence of it being a solid approach that  can be applied to general imitation learning in stochastic environments.
> - I disagree  with and discourage authors' attempt to present Hierarchy-NoPT as being universal representation of all existing works. It was also surprising to see that the authors tried to force the issue to avoid performing empirical comparisons with strong works. The positioning of the work is already very weak and not comparing it against strong baselines definitely makes the contributions very weak. In a future submission, I recommend the authors to include suggested baselines in empirical comparisons and also compare the limitations with their respective objective function and how they cannot handle conditional type shift.
>
> Finally, I agree with the other reviewer that the amount of changes in the revised paper are too many to be treated as the same paper. And even the revised version is not really improved technically or in presentation, which is still very cumbersome.  I hope that the authors can gain constructive insights from the review feedback and can work on carefully incorporating it in the future submission. However, the paper in its current form, is of very weak quality, and hence I cannot recommend it for acceptance.

---

### Official Review · Reviewer_4XDc · 2022-11-03

**Confidence:** 3
**Correctness:** 1
**Technical Novelty And Significance:** 2
**Empirical Novelty And Significance:** 2
**Recommendation:** 3

**Clarity, Quality, Novelty And Reproducibility:**

The writing quality and clarity are subpar; thus, it is hard to judge the originality of the work. In its current form, any proposed metrics or methods do not seem original. Even if it is a valid combination of existing approaches, that would be acceptable. But it is not explained in that way currently. It is still unclear why simple approaches would fail on their proposed setup.

**Strength And Weaknesses:**

## Strengths
- The overarching goal of replicating the entire distribution of an expert's behaviors in stochastic environments is well known to be important.
- Experiments are reported on a real-world dataset with good practical impact.

## Weaknesses
- **Writing and lack of clarity**: The paper is written in a way that is very hard to follow and understand the primary contributions of this work.
    + Many uncommon terms are used for existing alternate terminology, e.g., agent type (for trajectory encoding), hierarchical architecture (for variational methods), and distributional realism (which is already a goal of variational methods).
    + Lack of structure and flow across paragraphs makes it hard to follow.
        * The introduction lists many aspects and solutions but does not clearly list what the exact problem being solved in this paper is and how their contributions solve them.
        * Likewise, Section 3 is quite hard to follow. Is the paper solving two problems, marginal and conditional type shifts, or causal confusion as well?
    + The assumption of a known differentiable environment suddenly appears on Page 3 without any prior mention or justification.
    + What are the contributions of this paper's Robust Type Conditioning (RTC) method? From the writing, it seems that all 4 proposed losses are novel contributions. But that is undoubtedly not the case, as all of these losses are well-known and commonly used losses. The writing should be clear enough to distinguish what exists in the prior work and what is contributed by this method. This is also important for knowing what ablations should be performed. More on this is below.
- **Problem Formulation (Section 3)**:
    + A lot of issues are there with clarity.
        * If the majority of the problems in the paper come because of stochasticity from the environment in the input trajectory, then why is Figure 2 explained using a trajectory that has just one (s, a) pair? It makes it very hard to understand conditional-type shifts.
        * If marginal type shift is an existing problem in VAEs, called the prior hole problem, how is it solved in the prior work, and what is this paper doing differently?
- **Approach (Section 4)**:
    + Unclear why the problems mentioned in the paper arise when using standard approaches. For instance, training a standard state-conditioned (not trajectory-conditioned), variational GAIL policy should be able to resolve the problem of mode collapse, which is claimed to be one of the critical problems in the paper.
    + The paper says that the reconstruction loss prevents mode collapse. But, the paper also argues that the hierarchical (or variational) model is what prevents the mode collapse. The way it is written, it seems the reconstruction loss is a novelty. However, it is unclear to me how that is.
    + By minimizing the $L_2$ distance between agent true and predicted states at time $t$, how does the policy learn to imitate actions?
    + Why isn't there any KL term as is common in variational models? Would the $L_{prior}$ be needed if we have the standard KL loss?
    + The paper claims that $L_{adv}$ eliminates the marginal and conditional type shifts but does not explain how. Two paragraphs later, the paper claims that the conditional shift is not resolved, and thus information bottleneck is needed. Which one is right?
    + The claimed information bottleneck is not a novel solution and is well-known in the literature.
- **Experiments (Section 6)**:
    + Overall, because the paper's contributions are unclear, it is hard to understand what the ablations should be. In the current way of writing, I would imagine that all 4 losses should be ablated, but only "No Prior Training" is ablated.
    + While I appreciate the existing baselines and environments, it's hard to comment on the correctness without understanding the exact contributions of the paper.


**Summary Of The Paper:**

This paper aims to address the issue of replicating an expert's entire distribution of behaviors in imitation learning. In stochastic environments, an agent's behavior distribution can be caused due to its internal strategies or environment stochasticity. The paper claims that prior approaches suffer due to either not being distributional or because they do not disentangle environment stochasticity. This paper proposes existing tricks, such as using variational models, sampling from a prior distribution while training, and adding an information bottleneck to solve these issues.

**Summary Of The Review:**

To the best of my understanding, lack of clarity in writing, unjustified claims, and supposed lack of novelty are my reasons for recommending rejection.

---

> ### Author Response · Authors · 2022-11-18
> **Thank you!**
>
> We thank the reviewer for their helpful feedback.
>
> ## Highlights
>
> To start with, we would like to clarify that our two main contributions are:
> 1. Formulation of the conditional type shift and identification as an issue in stochastic environments
> 2. RTC as effective method to overcome this issue. The key novel algorithmic contribution is optimisation of the adversarial loss under prior conditioned types.
>
> We also rewrote section 3 and modified section 4 to improve clarity and better distinguish what is novel in our approach. We kindly ask you to have a look at the revised version and would be grateful for your feedback.
>
> Below, we will address your detailed points individually.
>
> ## Individual comments
>
> ### Writing and Lack of Clarity
>
> > Many uncommon terms are used for existing alternate terminology (“agent type” for trajectory encoding, “hierarchical architecture” for variational methods and “distributional realism”)
>
> * We believe that in the context of RL/IL, the term “hierarchical policy” is widespread.
> * We believe that using "trajectory encoding" instead of “agent type” would be misleading as it is not a trajectory encoding like in standard variational methods: It does not aim to encode the entire trajectory, but only the agent-internal factors!
> * We agree that distributional realism is a goal of variational methods, but it rarely is in imitation learning - and, as we show in this paper, tools from variational inference do not directly translate to stochastic environments due to the interactive nature of the IL problem setting, instead causing the issues outlined in section 3.
>
> > Lack of structure and flow across paragraphs makes it hard to follow.
> The introduction lists many aspects and solutions but does not clearly list what the exact problem being solved in this paper is and how their contributions solve them.
>
> We updated the "In this paper..." paragraph in the introduction to clearly summarise our contributions.
> As the problem definition itself is novel and requires all of section 3 to exactly define, we have to remain at a somewhat high level in the introduction.
>
>
> > The assumption of a known differentiable environment suddenly appears on Page 3 without any prior mention or justification.
>
> As described in section 4, RTC is applicable to both differentiable and non-differentiable environments.
> Our experiments are on differentiable environments as these are often used with autonomous vehicles (such as the Symphony Baseline on the WOMD dataset) - which we see as one of the most interesting applications of this work due to their high impact and high environment stochasticity.
>
>
> ### Problem Formulation
>
> > Why is Figure 2 explained using a trajectory that has just one (s, a) pair?
>
> This is the simplest model which can already include environmental stochasticity in state $s$. Having temporally extended trajectories, while more typical, greatly complicates the theoretical discussion as future states will not only depend on $\xi$, but also $g$.
> We added a sentence explaining that in the temporally extended situation, $\xi$ referers to all future stochasticity.
>
> > If marginal type shift is an existing problem in VAEs, called the prior hole problem, how is it solved in the prior work, and what is this paper doing differently?
>
> We agree that including it is not critical as the main novelty of our work lies in the conditional type shift and its solution through RTC. We have hence removed the discussion of the marginal type shift.
> To answer your question: Often it is not solved in practical applications of VAEs, but of course there exist various methods to tackle it - usually by making the prior more expressive, for example by using GMMs, flow-based encoders or importance-weighted samples.

---

> > ### Author Response · Authors · 2022-11-18
> > **Continuation of reply.**
> >
> > ### Approach (section 4)
> >
> > > Unclear why the problems mentioned in the paper arise when using standard approaches. For instance, training a standard state-conditioned (not trajectory-conditioned), variational GAIL policy should be able to resolve the problem of mode collapse, which is claimed to be one of the critical problems in the paper.
> >
> > We hope the improved section 3 describes the problem clearer.
> >
> > We are not sure what you are referring to with “standard state-conditioned variational GAIL”. GAIL is an adversarial method and while combinations with variational methods exist, we are unsure which one you are referring to.
> > * Only providing the current state would correspond to the standard GAIL policy - which we compare against in the form of the more powerful MGAIL policy and which exhibits mode collapse.
> > * Providing all future agent-states corresponds to “Hierarchy-NoPriorTraining (H-NoPT)”, which does not exhibit mode collapse but sufferers from the issues discussed in section 3.
> > * Providing only the next state one at a time would run into the same problems as providing all at once.
> >
> > > The paper says that the reconstruction loss prevents mode collapse.
> >
> > We agree that the formulation was misleading and updated it.
> >
> > > By minimizing the L2 distance between agent true and predicted states at time t, how does the policy learn to imitate actions?
> >
> > To arrive at the same point at time $t$, earlier actions $a_{< t}$ need to be similar.
> >
> > > Why isn't there any KL term as is common in variational models? Would the Lprior be needed if we have the standard KL loss?
> >
> > L_prior and L_ib together replace the standard KL loss from variational methods.
> > The standard KL loss cannot be applied in analytical form since we’re using a GMM for the prior for improved distributional realism.
> > By splitting it up into L_prior and L_ib, we can compute L_ib analytically while using samples for L_prior.
> >
> > > The paper claims that Ladv eliminates the marginal and conditional type shifts but does not explain how.
> >
> > We agree the formulation was confusing and updated the paragraphs. We hope this improves clarity and would kindly ask you to have another look.
> >
> > > The claimed information bottleneck is not a novel solution and is well-known in the literature.
> >
> > We agree and did not intend to make it seem so. We now moved the relevant citation to the first paragraph in section 4, hopefully making the relationship to prior literature clear.
> >
> > However, the bottleneck is an important component of the overall algorithm and we hence continue to mention its use in section 4. As we show in the experiments, only the combination of the information bottleneck with training under prior samples achieves both distributional realism and high task performance.
> >
> > ### Experiments
> >
> > > Overall, because the paper's contributions are unclear, it is hard to understand what the ablations should be.I n the current way of writing, I would imagine that all 4 losses should be ablated, but only "No Prior Training" is ablated.
> >
> > We hope the core contributions are clearer in the updated version of the paper. To re-state them here briefly:
> > * Identification of the conditional type shift as problem for hierarchical policies in stochastic environments
> > * The algorithm Robust Type Conditioning, for which the _novel_ algorithmic contribution is the training under prior-sampled types.
> >
> > Hence, our main ablation was the removed the prior-sampling of types.
> > However, for completeness, we now also added `RTC-NoIB` as additional ablation.
> > * The results for `RTC-NoIB` show that the bottleneck is necessary for RTC
> > * The results for `Hierarchy-NoPT` show that the bottleneck is insufficient for good performance.
> >
> > We believe ablating L_prior and L_adv by itself wouldn't be helpful as it is clear the algorithm couldn't work without them:
> > * Without L_prior, the prior would not be trained at all
> > * Without L_adv, the policy would not be trained at all under the prior (see also result for `Hierarchy-NoPT`)
> >
> > Ablating L_rec would be possible, but we're not sure it would be helpful to the reader as it is simply part of the standard variational method.
> >
> > > While I appreciate the existing baselines and environments, it's hard to comment on the correctness without understanding the exact contributions of the paper.
> >
> > We hope the contributions are now presented clearer in the paper.

---

> ### Comment · Reviewer_4XDc · 2022-12-09
> **Cannot recommend acceptance**
>
> Thank you for your rebuttal and edits to the paper. I tried to go through the edits, but it seems like a completely new paper now — with Sections 3 and 4 completely rewritten, including new theorems and proofs added.
> - The author response repeatedly refers to these changes in the paper, but I would have appreciated a **simple explanation of exactly what was changed and why**?
>     + These changes require a re-read of the entire paper. I cannot do that given that this is a rebuttal period, and a reasonable expectation is to treat it as a revision of the original paper — not a new submission.
> - I understand the changes are necessary based on the reviewers' feedback of lack of clarity and unsubstantiated claims. However, this magnitude of changes and the author's response acknowledge that there were **significant inconsistencies and incorrectness** in the original paper, including:
>     + The entire discussion and claims about **Marginal Type Shifts** as a problem being addressed in this paper.
>     + Reconstruction loss preventing mode collapse.
>     + **Sudden mention of GMM prior?** The original paper does not mention anything about the prior being GMM for improved distributional realism — which seems to be the basis for L_prior and L_ib, as per the author response. And even still, I don't understand why L_prior can be computed using samples from the GMM, and the standard KL loss cannot be? I am not denying this fact, but these are crucial pieces of information completely missing in the writing, and L_prior and L_ib are claimed as contributions for totally different reasons.
>     + The paper claims that Ladv eliminates the marginal and conditional type shifts but does not explain how.
>     + Information bottleneck claimed as a novel solution instead of being justified as an important component for the environments considered.
> - I still don't see a simple, clear, and concise definition of problem and contribution. The introduction writing is still quite subpar in making the paper easy to follow. Maybe in a future version of the paper, the authors could rewrite a concise intro, simply stating (and not just explaining) what the problem is, what the prior methods cannot do, and how this paper solves the problem.
>
>
>
> All in all, the original submission was far from a good paper quality because of incorrectness and lack of clarity. The updated version is potentially better, but it still (a) does not address the clarity issues properly, and (b) hard to judge its reliability as I have no idea what the changes are with respect to the original version and whether more incorrect/unjustified claims are made.
>
> **Note to Authors**: I appreciate the attempt at an important problem with a real-world dataset, but please understand that the paper is not possible to accept as a reviewer in its current form. I hope my original review and the above feedback are helpful for the next iteration of this paper, while I admit my limitations in being able to understand the paper fully because of its writing. I do believe a thorough rewriting and restructuring could enable this paper to be of acceptable quality in the future.

---

### Decision · Program_Chairs · 2023-01-20

**Decision:**

Reject

**Justification For Why Not Higher Score:**

Lack of clear exposition. Limited experimental evidence.

**Justification For Why Not Lower Score:**

N/A

**Metareview: Summary, Strengths And Weaknesses:**

The paper proposes a new algorithm for imitation learning in stochastic environments. This setting is generally challenging as many standard algorithms can conflate (expert) policy noise with environment noise when learning from demonstrations. The algorithm combines an adversarial and variational formulation, along with an information bottleneck based regularization. Empirically, the algorithm outperforms MGAIL and InfoMGAIL on a driving environment.

On the positive side, the reviewers appreciated the motivation to address unique challenges in stochastic environments, experiments with a realistic driving environment, and some of the fine-grained definitions and metrics used for evaluation of the framework. However, the reviewers also had many concerns. First, the reviewers generally found the exposition unclear --- this was complicated by the fact that almost entire sections were rewritten post rebuttal and one challenge (marginal type shift) removed entirely. Second, the reviewers wanted to see some more modern baselines that also explicitly tackle learning of multimodal policies via imitation. Finally, given the multiple terms in the loss function, the reviewers wanted to see a more detailed ablation of the different terms and their coefficients, as well as tuning/sensitivity of the additional hyperparameters.